# Improving the Generation and Evaluation of Synthetic Data for Downstream Medical Causal Inference

**Harry Amad**
DAMTP
University of Cambridge
hmka3@cam.ac.uk

**Zhaozhi Qian**[*]
Elm Europe
zqian@elm.sa

**Dennis Frauen**
LMU Munich
Munich Center for Machine Learning

**Julianna Piskorz**
DAMTP
University of Cambridge

**Stefan Feuerriegel**
LMU Munich
Munich Center for Machine Learning

**Mihaela van der Schaar**
DAMTP
University of Cambridge

## Abstract

Causal inference is essential for developing and evaluating medical interventions, yet real-world medical datasets are often difficult to access due to regulatory barriers. This makes synthetic data a potentially valuable asset that enables these medical analyses, along with the development of new inference methods themselves. Generative models can produce synthetic data that closely approximate real data distributions, yet existing methods do not consider the unique challenges that downstream causal inference tasks, and specifically those focused on *treatments*, pose. We establish a set of desiderata that synthetic data containing treatments should satisfy to maximise downstream utility: preservation of (i) the covariate distribution, (ii) the treatment assignment mechanism, and (iii) the outcome generation mechanism. Based on these desiderata, we propose a set of evaluation metrics to assess such synthetic data. Finally, we present STEAM: a novel method for generating *Synthetic data for Treatment Effect Analysis in Medicine* that mimics the data-generating process of data containing treatments and optimises for our desiderata. We empirically demonstrate that STEAM achieves state-of-the-art performance across our metrics as compared to existing generative models, particularly as the complexity of the true data-generating process increases.

## 1 Introduction

Access to medical data is crucial for advancing healthcare research, enabling novel medical discoveries and providing a testbed for developing new analytical approaches, such as machine-learning-based causal inference methods [6, 94, 28]. However, regulations restrict access to patient data for research purposes [3, 92]. Synthetic data, which is gaining significant recognition in medical literature [48], offers a way to increase data availability, and recent initiatives are aiming to generate synthetic data for broad 'health data research' [20, 72], such as testing of learning algorithms [33] and replication of clinical trial results [17]. Importantly, the promise of synthetic data hinges on its ability to preserve information critical to relevant downstream tasks. The majority of recent literature [7] focuses on downstream *predictive* tasks, shaping standard evaluation and generation practices to this setting.

Medical data typically contain *treatment assignment variables* which invite unique approaches to downstream analysis, beyond standard prediction [28]. Data containing treatments are typically analysed via *causal inference* (e.g. treatment effect estimation methods) to examine causal relationships

---

[*]Work completed while at University of Cambridge.

39th Conference on Neural Information Processing Systems (NeurIPS 2025).

between covariates, treatments, and outcomes. Despite this, many works in synthetic data which use medical data for motivation and validation [13, 58, 98, 10] employ standard, prediction-oriented, generation and evaluation techniques (see Appendix A for more details). Such failure to acknowledge the likely downstream uses of synthetic data containing treatments leads to low-quality data generation, and evaluation via misaligned metrics.

**Evaluation.** Standard synthetic data evaluation involves holistic statistical comparisons of synthetic and real data (Table 1). In this paradigm, causal inference tasks are overlooked, as treatments are handled like any other variable, limiting the relevance of such assessment in medical settings. To illustrate this, consider the following key questions that an analyst working with a synthetic medical dataset, $\mathcal{D}_s$, may ask: **Q1** *How representative are the patient covariates in* $\mathcal{D}_s$*?*; **Q2** *How accurate are the treatment assignment decisions in* $\mathcal{D}_s$*?*; and **Q3** *How much error might be introduced in treatment effect estimates derived from* $\mathcal{D}_s$*?* These questions require differentiation between covariates, treatments, and outcomes, and they cannot be answered with current evaluation protocols.

**Generation.** *Generic* generative models (Table 2) are typically designed to minimise the divergence between real and synthetic joint distributions. In doing so, these models overlook the specific causal relationships between covariates, treatments, and outcomes that are essential for downstream medical applications. Existing *causal* generative models, on the other hand, generally assume access to a causal graph that describes all causal relationships between variables, which is often unrealistic in complex domains like medicine, where true causal structure is rarely known [27].

In this work we address these limitations by proposing novel approaches to evaluation and generation of synthetic data containing treatments, operating under reasonable assumptions and explicitly considering the downstream use of such data. In doing so, we make the following contributions:

1. **Desiderata:** By examining typical medical analyses conducted on data containing treatments, we establish a set of desiderata that synthetic data should satisfy in this context (§4).

2. **Evaluation:** We show that existing synthetic data evaluation metrics are inadequate in this setting, as they do not measure adherence to these desiderata. As a remedy, we propose a principled set of metrics derived from our desiderata, allowing meaningful evaluation of synthetic data containing treatments (§5).

3. **Generation:** We propose STEAM, a method for generating synthetic data that augments generic generative models to encode inductive biases to optimise for our desiderata (§6).

4. **Empirical analysis:** We demonstrate that STEAM generates state-of-the-art synthetic data containing treatments, particularly as the real data-generating process (DGP) grows in complexity, and in high-dimensional scenarios (§7).[2]

## 2  Problem formulation

**Setup.** We consider a *data owner* with access to observational or experimental real data $\mathcal{D}_r = \{(\mathbf{X}_r^{(i)}, W_r^{(i)}, Y_r^{(i)})\}_{i=1}^n$ sampled from a population $P$, where $\mathbf{X}_r^{(i)} \in \mathcal{X}$ is a vector of $d$ binary or continuous covariates, $W_r^{(i)} \in \{0,1\}$ is a binary treatment assignment, and $Y_r^{(i)} \in \mathcal{Y}$ is a binary or continuous outcome. We refer to the set of all variables in $\mathcal{D}_r$ as $\mathcal{V} = \{X_1, ..., X_d, W, Y\}$. We denote the propensity score with $\pi(\mathbf{x}) = P_{W|\mathbf{X}}(W = 1 \mid \mathbf{X} = \mathbf{x})$.

**Objective.** We wish to enable the release of synthetic data to downstream users with various analysis goals, such as estimation of propensity scores, average treatment effects (ATEs), and conditional average treatment effects (CATEs). To do so, we aim to generate synthetic data $\mathcal{D}_s = \{(\mathbf{X}_s^{(i)}, W_s^{(i)}, Y_s^{(i)})\}_{i=1}^n$ from a distribution $Q$ and evaluate how well $\mathcal{D}_s$ captures the information in $\mathcal{D}_r$ that is relevant to likely downstream tasks with a set of metrics $\mathcal{M}(\mathcal{D}_r, \mathcal{D}_s)$.

## 3  Related work

**Evaluation.** Evaluation of synthetic tabular data, the modality we focus on here, has two common focuses: *resemblance* and *predictive utility* [73]. While *predictive utility* assesses model accuracy when

---

[2]Our code is available at `https://github.com/harrya32/STEAM` and `https://github.com/vanderschaarlab/STEAM`.

Table 1: Synthetic data evaluation methods. $d$ and $\rho$ are distance and correlation functions, $\mathcal{S}_P$ is the support of distribution $P$. '**Q?**': which of the questions from §1 does the method answer?

| | Method | Formula | Differentiates between $\mathbf{X}, W, Y$? | Q? |
|---|---|---|---|---|
| *Existing* | Marginal | $\frac{1}{|\mathcal{V}|}\sum_{i\in\mathcal{V}}d(P_i,Q_i)$ | ✗ | — |
| | Correlation | $\frac{1}{|\mathcal{V}|(|\mathcal{V}|-1)}\sum_{\substack{i,j\in\mathcal{V}\\i\neq j}}d(\rho(P_i,P_j),\rho(Q_i,Q_j))$ | ✗ | — |
| | Joint | $d(P_\mathcal{V},Q_\mathcal{V})$ | ✗ | — |
| | Prec., Rec. | $|\mathcal{S}_P\cap\mathcal{S}_Q|/|\mathcal{S}_Q|, |\mathcal{S}_P\cap\mathcal{S}_Q|/|\mathcal{S}_P|$ | ✗ | — |
| *Ours* | $P_{\alpha,\mathbf{X}}, R_{\beta,\mathbf{X}}$ | Equations 3 and 4 | ✓ | *Q1* |
| | $\mathrm{JSD}_\pi$ | Equation 5 | ✓ | *Q2* |
| | $U_{\mathrm{PEHE}}$ | Equation 6 | ✓ | *Q3* |

Table 2: Synthetic data generation methods. $d$ is a distance function. $\mathrm{PA}_\mathcal{G}(\mathcal{V}_i)$ refers to the set of parents of node $\mathcal{V}_i$ in the causal graph $\mathcal{G}$.

| | Example methods | Distributional target | Assumptions on $\mathcal{D}_\mathbf{r}$ | $\mathcal{D}_\mathbf{s}$ application |
|---|---|---|---|---|
| *Generic gen. models* | NFlow [80]
CTGAN [97]
TVAE [97]
TabDDPM [58]
ARF [93] | $\min d(Q_\mathcal{V},P_\mathcal{V})$ | None | Prediction |
| *CGMs* | DCM [12]
VACA [83] | $\min d\big(Q_{\mathcal{V}_1|\mathrm{PA}_\mathcal{G}(\mathcal{V}_1)},P_{\mathcal{V}_1|\mathrm{PA}_\mathcal{G}(\mathcal{V}_1)}\big)$
$\vdots$
$\min d\big(Q_{\mathcal{V}_{|\mathcal{V}|}|\mathrm{PA}_\mathcal{G}(\mathcal{V}_{|\mathcal{V}|})},P_{\mathcal{V}_{|\mathcal{V}|}|\mathrm{PA}_\mathcal{G}(\mathcal{V}_{|\mathcal{V}|})}\big)$ | $\mathcal{G}$ | Interventional/
counterfactual
queries on $\mathcal{G}$ |
| *Ours* | STEAM | $\min d(Q_\mathbf{X},P_\mathbf{X})$
$\min d(Q_{W|\mathbf{X}},P_{W|\mathbf{X}})$
$\min d(Q_{Y|W,\mathbf{X}},P_{Y|W,\mathbf{X}})$ | Valid DGP | Caual
inference with
treatments |

trained on $\mathcal{D}_\mathrm{s}$, it is unsuitable for causal tasks lacking observable ground-truths [44]. On the other hand, *resemblance* metrics compare statistical properties of $\mathcal{D}_\mathrm{r}$ and $\mathcal{D}_\mathrm{s}$ (marginals, correlations, joint distributions, supports; Table 1). Existing methods handle all variables similarly, not differentiating between $\mathbf{X}$, $W$, and $Y$, and they therefore cannot answer *Q1-3*. Our metrics (§5) remedy this.

**Generation.** We consider two related approaches to generative modeling (Table 2). Firstly, *generic generative models* make minimal assumptions on $\mathcal{D}_\mathrm{r}$, minimising the difference between real and synthetic joint distributions. In doing so, these methods forego any treatment-related inductive biases, unlike our STEAM method. *Causal generative models (CGMs)*, on the other hand, require full knowledge of the causal graph, $\mathcal{G}$, to model each conditional distribution as dictated by $\mathcal{G}$. STEAM's assumptions (§6), that the underlying DGP of $\mathcal{D}_\mathrm{r}$ is of the form $\mathbf{X} \sim P_\mathbf{X}$, $W \sim P_{W|\mathbf{X}}$, $Y \sim P_{Y|W,\mathbf{X}}$, are much less restrictive, not requiring knowledge of individual causal relationships between variables, and they will hold for a wide array of datasets containing treatments. For further elaboration on specific evaluation and generation methods, see Appendix B.

## 4 Desiderata for synthetic data containing treatments

We cover three essential distributions for treatment data: (i) the covariate distribution $P_\mathbf{X}$, (ii) the treatment assignment mechanism $P_{W|\mathbf{X}}$, and (iii) the outcome generation mechanism $P_{Y|W,\mathbf{X}}$. While their importance is clear in the causal inference community, hence *Q1-3*, this has not been adequately recognised by the synthetic data community, and methods that target them are missing. To bridge this gap, we establish desiderata for synthetic data containing treatments based on these distributions.

**(i) The covariate distribution $P_{\mathbf{X}}$** describes the population of interest and, in medicine, it is standard to report its characteristics [95], as it determines to whom analysis is relevant.

*Why is its preservation important?* Inadequate covariate coverage in $\mathcal{D}_s$ can result in exclusion from downstream analysis of members of the population whose covariates are not well explored, as making reliable inferences can become infeasible [78, 81]. On the other hand, generating out-of-distribution covariates in $\mathcal{D}_s$ can cause groundless extrapolation by synthetically-trained models, leading to potential misuse.

**(ii) The treatment assignment mechanism $P_{W|\mathbf{X}}$** is a nuisance parameter in many treatment effect models [4, 15], and it can be a target for analysis itself when examining treatment protocols.

*Why is its preservation important?* Since $P_{W|\mathbf{X}}$ is used as a nuisance parameter, errors in its modelling propagate to treatment effect estimates derived from $\mathcal{D}_s$. Furthermore, $P_{W|\mathbf{X}}$ can guide the difficult task of CATE model selection [47], so poor preservation can lead to inconsistency in model selection between $\mathcal{D}_r$ and $\mathcal{D}_s$, which is unideal [37]. Misrepresenting $P_{W|\mathbf{X}}$ also risks misreporting treatment protocols. Given that extreme propensities of $\pi(\mathbf{x}) \approx 0$ (or $\pi(\mathbf{x}) \approx 1$) are common in data such as electronic health records [63], often because of safety, inaccurate $Q_{W|\mathbf{X}}$ could encourage exploration of treatments in patient subgroups for which they are highly unsafe.

**(iii) The outcome generation mechanism $P_{Y|W,\mathbf{X}}$** is the distribution through which treatment effects can be estimated, by comparing the statistical functionals of $P_{Y|W=1,\mathbf{X}}$ and $P_{Y|W=0,\mathbf{X}}$.

*Why is its preservation important?* $P_{Y|W,\mathbf{X}}$ must be preserved, so that $\mathcal{D}_s$ can permit accurate estimation of treatment effects. If $Q_{Y|W,\mathbf{X}}$ is inaccurate, then even a perfect model could not estimate correct treatment effects from $\mathcal{D}_s$, and the worse this relationship is preserved, the less useful $\mathcal{D}_s$ becomes.

Preserving (i)–(iii) is *necessary* and *sufficient* for $Q$ to be a high-quality approximation of $P$. Modelling each distribution well is evidently *necessary* given the above reasons, and it is also *sufficient*, which is clear from the following decomposition of $P$:

$$P(\mathbf{X}, W, Y) = \underbrace{P_{\mathbf{X}}(\mathbf{X})}_{(i)} \underbrace{P_{W|\mathbf{X}}(W|\mathbf{X})}_{(ii)} \underbrace{P_{Y|W,\mathbf{X}}(Y|W,\mathbf{X})}_{(iii)} \tag{1}$$

The components (i)–(iii) offer a complete factorisation of the joint distribution, and therefore $Q$ matching $P$ in each component is sufficient for $Q$ to match $P$ entirely. As such, accurate modelling of (i)–(iii) forms our desiderata for synthetic data containing treatments. Generation methods should seek to *maximise adherence to these desiderata*, and evaluation metrics should *assess how successful $\mathcal{D}_s$ is in this regard* (and therefore answer **Q1-3**).

**On causal assumptions.** Even if these desiderata are satisfied, $\mathcal{D}_s$ may not permit useful analysis via causal inference. Assumptions, such as typical identifiability assumptions,[3] must still be critically examined by analysts, since any violations in $\mathcal{D}_r$ will almost surely be violated in a faithful $\mathcal{D}_s$ as well. Accounting for violated assumptions is a task orthogonal to synthetic data generation, with existing literature [52, 29], and we do not consider it necessary for $Q$ to improve upon any biases in $P$, allowing post-generation methods to rectify them, if necessary, instead.

## 5 How to evaluate synthetic data containing treatments

With our desiderata established, we now investigate how to evaluate the adherence of $\mathcal{D}_s$.

### 5.1 Inadequacy of existing metrics

Existing evaluation metrics do not measure how well $\mathcal{D}_s$ satisfies our desiderata. These metrics do not differentiate between $\mathbf{X}$, $W$, and $Y$, and they therefore cannot directly assess any of $Q_{\mathbf{X}}$, $Q_{W|\mathbf{X}}$, or $Q_{Y|W,\mathbf{X}}$. Joint-distribution-level metrics, such as Kullback–Leibler (KL) divergence [59], are the most common approach, offering a holistic assessment of how well $Q$ models $P$. However, these are

---

[3]Consistency: $Y^{(i)} = Y(W^{(i)})$, overlap: $0 < \pi(\mathbf{x}) < 1$, and unconfoundedness: $Y(0), Y(1) \perp\!\!\!\perp W|\mathbf{X}$

only loosely related to our desiderata, and they do not allow a user to disentangle how each of (i)–(iii) is preserved, limiting the depth of information offered on $\mathcal{D}_s$. Furthermore, we argue that they tend to be dominated by $\mathbf{X}$ as it grows in dimensionality, and they will lose *sensitivity* to the treatment assignment and outcome generation mechanisms. By this, we mean that these metrics tend to fail to notice differences in the modelling of $P_{W|\mathbf{X}}$ or $P_{Y|W,\mathbf{X}}$ by two proposal distributions.

For an illustrative example of this phenomenon with KL divergence, consider a simple $P$ which can be factorized as $P = \prod_{i=1}^{d} P_{\mathbf{X}_i} \, P_{W|\mathbf{X}} \, P_{Y|W,\mathbf{X}}$. Let there be two learnable distributions $Q^{\theta_1}$ and $Q^{\theta_2}$ with the same form $Q^{\theta_k} = \prod_{i=1}^{d} Q_{\mathbf{X}_i}^{\theta_{\mathbf{X}}} \, Q_{W|\mathbf{X}}^{\theta_{W,k}} \, Q_{Y|W,\mathbf{X}}^{\theta_{Y,k}}$, and which only differ in either $\theta_{W,k}$ or $\theta_{Y,k}$ (i.e., they either model $P_{W|\mathbf{X}}$ or $P_{Y|W,\mathbf{X}}$ differently). In this setting, the following holds:

**Theorem 1.** *Let $P$, $Q^{\theta_1}$, $Q^{\theta_2}$ be of the above form, and $\mathcal{M}$ be KL divergence. If we assume that $Q^{\theta_1}$ and $Q^{\theta_2}$ have sufficient capacity to have bounded error on each component, i.e. $\forall i$, $0 < \mathcal{M}(P_{\mathbf{X}_i}, Q_{\mathbf{X}_i}^{\theta_{\mathbf{X}}}) < \varepsilon_{\mathbf{X}}$, and $0 < \mathcal{M}(P_{W|\mathbf{X}}, Q_{W|\mathbf{X}}^{\theta_{W,k}}) < \varepsilon_{W,k}$, and $0 < \mathcal{M}(P_{Y|W,\mathbf{X}}, Q_{Y|W,\mathbf{X}}^{\theta_{Y,k}}) < \varepsilon_{Y,k}$, then:*

$$\frac{\mathcal{M}(P, Q^{\theta_1})}{\mathcal{M}(P, Q^{\theta_2})} \to 1, \; as \; d \to \infty \tag{2}$$

*Proof.* See Appendix C. $\qquad\qquad\qquad\qquad\qquad\qquad\qquad\qquad\qquad\qquad\qquad\qquad\square$

Theorem 1 shows that KL divergence loses sensitivity to $W|\mathbf{X}$ and $Y|W,\mathbf{X}$ as $d$ grows, suggesting that it will struggle to select between $Q^{\theta_1}$ and $Q^{\theta_2}$, despite any difference in their modelling of $P_{W|\mathbf{X}}$ or $P_{Y|W,\mathbf{X}}$. For an empirical example of this phenomenon arising across an extended array of joint-distribution-level metrics, see Appendix D.

## 5.2 Metrics for synthetic data containing treatments

These findings motivate us to design our own metrics for synthetic data containing treatments which directly measure adherence to desiderata (i)–(iii) and offer answers to ***Q1-3***.

### 5.2.1 The covariate distribution

Evaluating $Q_{\mathbf{X}}$ requires comparing the generally high-dimensional covariate distributions of $\mathcal{D}_r$ and $\mathcal{D}_s$, which is non-trivial. Nevertheless, this is a standard synthetic data evaluation task, as $\mathbf{X}$ does not contain treatments. We see precision/recall analysis as the most useful approach. There is typically a trade-off between these two qualities, which generative models balance differently [8, 82], and by measuring them both, a data holder can guide generation towards their preferences for covariate realism and diversity. Without a strong preference, balancing the two is recommended [50].

We propose the use of the integrated $P_\alpha$ and $R_\beta$ scores, introduced in [2], which compare the $\alpha$-supports of $\mathcal{D}_r$ and $\mathcal{D}_s$, for $\alpha \in [0, 1]$.[4] Intuitively, $P_\alpha$ captures how well $\mathcal{D}_s$ falls within the support of $\mathcal{D}_r$, and $R_\beta$ reflects how well $\mathcal{D}_r$ is covered by the support of $\mathcal{D}_s$. We denote the covariate precision and recall with $P_{\alpha,\mathbf{X}}$ and $R_{\beta,\mathbf{X}}$ respectively, which are calculated by applying integrated $P_\alpha$ and $R_\beta$ to the covariates of $\mathcal{D}_r$ and $\mathcal{D}_s$ only, as in Eq. (3) and Eq. (4).

$$P_{\alpha,\mathbf{X}}(\mathcal{D}_r, \mathcal{D}_s) = 1 - 2 \int_0^1 |\mathbb{P}(\tilde{\mathbf{X}}_s \in \mathcal{S}_r^\alpha) - \alpha| \, d\alpha \tag{3}$$

$$R_{\beta,\mathbf{X}}(\mathcal{D}_r, \mathcal{D}_s) = 1 - 2 \int_0^1 |\mathbb{P}(\tilde{\mathbf{X}}_r \in \mathcal{S}_s^\beta) - \beta| \, d\beta \tag{4}$$

where $\tilde{\mathbf{X}}_\diamond$ and $\mathcal{S}_\diamond^\square$ are the embedding $\tilde{\mathbf{X}}_\diamond = \Phi(\mathbf{X}_\diamond)$ and $\square$-support as defined in [2], respectively.

We have $0 < P_{\alpha,\mathbf{X}}, R_{\beta,\mathbf{X}} < 1$, and scores near 1 indicate a realistic and diverse $Q_{\mathbf{X}}$. Together, these metrics can be used to answer ***Q1***.

---

[4]An $\alpha$-support is the minimum volume subset of the domain that contains probability mass $\alpha$.

### 5.2.2 The treatment assignment mechanism

While in general we do not have access to $P_{W|\mathbf{X}}$ and $Q_{W|\mathbf{X}}$, we know that, for each $\mathbf{X} = \mathbf{x}$, they are Bernoulli distributions, since $W$ is a binary variable. The success probabilities can be estimated from $\mathcal{D}_r$ and $\mathcal{D}_s$ with a probabilistic classifier, which can be used to form approximations of $P_{W|\mathbf{X}}$ and $Q_{W|\mathbf{X}}$. There is then an array of valid options to compare these approximations. We propose the use of Jensen-Shannon distance,[5] given its desirable properties of symmetry, smoothness, and boundedness (we discuss alternatives in Appendix E). For a given probabilistic classifier $\hat{\pi}$, we define $\hat{P}_{W|\mathbf{X}=\mathbf{x}} = \mathrm{Bern}(\hat{\pi}_r(\mathbf{x}))$ and $\hat{Q}_{W|\mathbf{X}=\mathbf{x}} = \mathrm{Bern}(\hat{\pi}_s(\mathbf{x}))$ where $\hat{\pi}_r$ and $\hat{\pi}_s$ are trained on $\mathcal{D}_r$ and $\mathcal{D}_s$ respectively, and we measure the preservation of $P_{W|\mathbf{X}}$ as in Eq. (5).

$$\mathrm{JSD}_\pi(\mathcal{D}_r, \mathcal{D}_s) = 1 - \mathbb{E}_{P_\mathbf{X}}\left[\sqrt{\frac{1}{2}(\mathrm{KL}(\hat{P}_{W|\mathbf{X}=\mathbf{x}} \,\|\, M) + \mathrm{KL}(\hat{Q}_{W|\mathbf{X}=\mathbf{x}} \,\|\, M))}\right] \tag{5}$$

where $M = \frac{1}{2}(\hat{P}_{W|\mathbf{X}=\mathbf{x}} + \hat{Q}_{W|\mathbf{X}=\mathbf{x}})$ and KL is KL divergence using $\log_2$.

$\mathrm{JSD}_\pi$ can be used to answer **Q2**. We have $0 < \mathrm{JSD}_\pi < 1$, with scores near 1 indicating that $Q_{W|\mathbf{X}}$ matches $P_{W|\mathbf{X}}$ well. The validity of $\mathrm{JSD}_\pi$ will depend on the accuracy of $\hat{\pi}$, so conducting $\hat{\pi}$ model selection is an important pre-evaluation step, although, amongst reasonable model choices, we find the information offered by $\mathrm{JSD}_\pi$ does not significantly differ.

### 5.2.3 The outcome generation mechanism

To evaluate the preservation of $P_{Y|W,\mathbf{X}}$, we consider a treatment effect analogue of predictive utility. We address the unavailability of ground-truths by seeking parity in performance between $\mathcal{D}_r$ and $\mathcal{D}_s$, rather than quantifying error from an oracle value. Such evaluation is inherently task dependent, yet the specific quantity $\mathcal{D}_s$ may be used to estimate is unclear. Assessment should therefore centre on a complex task, in which comparable performance will likely imply the same for simpler tasks. In this case, we consider the most difficult treatment effect task likely to arise in the medical field—CATE estimation—as similarity in this between $\mathcal{D}_s$ and $\mathcal{D}_r$ will tend to imply similarity in simpler tasks, such as ATE estimation.[6] If $\mathcal{D}_s$ yields accurate CATEs across the full patient population then, by definition, it will also yield an accurate ATE, but the reverse is not true, i.e., $\mathcal{D}_s$ can reproduce the correct ATE and yet contain arbitrarily incorrect CATEs. Therefore, we evaluate how well $Q_{Y|W,\mathbf{X}}$ preserves $P_{Y|W,\mathbf{X}}$ by calculating the PEHE between synthetic- and real-trained CATE learners (see Appendix E for alternatives). Given a family $\mathcal{F}$ of CATE learners $\hat{\tau}$, where $\hat{\tau}_r$ and $\hat{\tau}_s$ are trained on $\mathcal{D}_r$ and $\mathcal{D}_s$ respectively, we assess the preservation of $P_{Y|W,\mathbf{X}}$ as in Eq. (6).

$$U_{\mathrm{PEHE}}(\mathcal{D}_r, \mathcal{D}_s) = \frac{1}{|\mathcal{F}|} \sum_{\hat{\tau} \in \mathcal{F}} \sqrt{\mathbb{E}_{P_\mathbf{X}}[(\hat{\tau}_s(\mathbf{X}) - \hat{\tau}_r(\mathbf{X}))^2]} \tag{6}$$

$U_{\mathrm{PEHE}}$ can answer **Q3**. We average over $\mathcal{F}$ since CATE model validation is difficult [16], so $\hat{\tau}$ cannot be set as the best performing model in a similar fashion as is done for $\mathrm{JSD}_\pi$ (we discuss choices for $\mathcal{F}$ in Appendix F). As such, $U_{\mathrm{PEHE}}$ rewards synthetic data which permit proximity in CATEs across an array of potential learners, where a lower $U_{\mathrm{PEHE}}$ indicates better preservation of $P_{Y|W,\mathbf{X}}$.

## 6  Generating synthetic data containing treatments

To illustrate the standard DGP of data containing treatments, shown in the middle of Figure 4, consider a simple hospital dataset. Patient covariates $\mathbf{X}$, such as height, weight, etc., are drawn from an underlying covariate distribution $P_\mathbf{X}$, which is dictated by the local population. Treatments are then assigned by a domain expert, such as a doctor, conditioned on $\mathbf{X}$, i.e., $W \sim P_{W|\mathbf{X}}$. Finally, patients' outcomes are dictated by the dynamics of their ailments, conditional upon $W$ and $\mathbf{X}$, i.e., $Y \sim P_{Y|\mathbf{X},W}$. We now propose STEAM, a novel model-agnostic framework for generating *Synthetic data for Treatment Effect Analysis in Medicine* which mimics this DGP.

---

[5]$\mathrm{JSD}(P \,\|\, Q) = \sqrt{\frac{1}{2}\mathrm{KL}(P \,\|\, M) + \frac{1}{2}\mathrm{KL}(Q \,\|\, M)}$, where $M = \frac{1}{2}(P + Q)$.

[6]Denoting the potential outcomes as Y(0) and Y(1), ATE is defined as $\mathrm{ATE} = \mathbb{E}_P[Y(1) - Y(0)]$ and CATE is $\tau(\mathbf{x}) = \mathbb{E}_P[Y(1) - Y(0)|\mathbf{X} = \mathbf{x}]$.

## 6.1 STEAM

STEAM, shown on the right of Figure 4, conducts a three-step generation process, mimicking the real DGP to push $Q$ closer towards $P$ in structure and directly target each distribution from our desiderata. STEAM involves three components:

1. $Q_{\mathbf{X}}$. $\mathbf{X}$ is generated from a generic generative model trained to match $P_{\mathbf{X}}$.

2. $Q_{W|\mathbf{X}}$. Treatments are assigned according to a propensity function trained on $\mathcal{D}_{\mathrm{r}}$. If $\mathcal{D}_{\mathrm{r}}$ is experimental data with known $P_{W|\mathbf{X}}$, then $Q_{W|\mathbf{X}}$ can be directly set as the true distribution, negating the need for any optimisation at this step.

3. $Q_{Y|W,\mathbf{X}}$. Potential outcome (PO) estimators are trained on $\mathcal{D}_{\mathrm{r}}$ to match $P_{Y|W=0,\mathbf{X}}$ and $P_{Y|W=1,\mathbf{X}}$, and the relevant outcome is generated for each instance based on their assigned treatment.

Each component can be defined from a diverse array of potential models. $Q_{\mathbf{X}}$ can be any generic generative model, $Q_{W|\mathbf{X}}$ can be any classifier, and $Q_{Y|W,\mathbf{X}}$ can use any regressors. Generation via the STEAM framework can therefore be framed as augmenting *any* generic base generative model to improve its generation of medical data for use in downstream causal inference tasks, allowing it to easily fit within existing synthetic data generation pipelines.

## 7 Empirical analysis

In §7.1.1 and §7.1.2, we compare generic generative models and CGMs, respectively, with STEAM models on medical data. Then, in §7.2, we examine performance in a number of targeted settings to better understand where the STEAM framework is particularly successful.

In STEAM, we consistently set $Q_{W|\mathbf{X}}$ as a logistic regression classifier, and $Q_{Y|W,\mathbf{X}}$ as the PO estimators from S-learner [60]. We use the open source `synthcity` [79] for all generic generative models, and we indicate what we set for $Q_{\mathbf{X}}$ in STEAM with subscript, i.e., STEAM$_\diamond$ uses generative model $\diamond$ for $Q_{\mathbf{X}}$. We detail data and experimental set-ups in Appendix H.

### 7.1 Generation of medical data containing treatments

To compare STEAM with existing generation frameworks, we consider performance across three medical datasets:

1. **AIDS Clinical Trial Group (ACTG) study 175.** A trial on subjects with HIV-1 [36].

2. **Infant Health and Development Program (IHDP).** A semi-synthetic medical dataset, with real covariates and simulated outcomes, using data from an experiment evaluating the effect of specialist childcare on the cognitive scores of premature infants [11].

3. **Atlantic Causal Inference Competition 2016 (ACIC).** A semi-synthetic medical dataset, with real covariates and simulated outcomes, containing data from the Collaborative Perinatal Project [74].

### 7.1.1 Comparison with generic generative models

We compare state-of-the-art generic tabular data generators with their STEAM counterparts across all three datasets. We choose baselines across the major families of tabular data generators: CTGAN [97], TVAE [97], ARF [93], NFlow [80], and TabDDPM [58]. We display comprehensive results for each dataset and model combination in Table 3.

> **Takeaway.** In the vast majority of cases, generation via STEAM leads to better performance across our metrics. Improvements in terms of $\mathrm{JSD}_\pi$ and $U_{\mathrm{PEHE}}$ are most notable, indicating that STEAM significantly increases the preservation of $P_{W|\mathbf{X}}$ and $P_{Y|W,\mathbf{X}}$. There is relatively little difference in $P_{\alpha,\mathbf{X}}$ and $R_{\beta,\mathbf{X}}$ between generic and STEAM models, which is expected. While STEAM does isolate the modelling of the covariates in $Q_{\mathbf{X}}$, giving that component model an ostensibly easier task than its joint-level alternative, modelling the complete $P_{\mathbf{X},W,Y}$ is nearly of equivalent difficulty to modelling $P_{\mathbf{X}}$ when the number of covariates is high, since they dominate the dimensionality. Table 3 also demonstrates that STEAM's performance is sensitive to the

Table 3: Medical data generation with generic and STEAM models, averaged over 20 runs with 95% CIs. Coloured numbers are the relative differences between each STEAM and generic model.

| Dataset | Model | $P_{\alpha,\mathbf{x}}$ (↑) | $R_{\beta,\mathbf{x}}$ (↑) | $\mathrm{JSD}_{\pi}$ (↑) | $U_{\mathrm{PEHE}}$ (↓) |
|---|---|---|---|---|---|
| ACTG | TVAE | $0.926 \pm 0.013$ | $0.483 \pm 0.010$ | $0.946 \pm 0.004$ | $0.564 \pm 0.017$ |
| | STEAM $_{\mathrm{TVAE}}$ | $0.929 \pm 0.008$ (+0.003) | $0.486 \pm 0.009$ (+0.003) | $0.958 \pm 0.004$ (+0.012) | $0.492 \pm 0.011$ (-0.072) |
| | ARF | $0.818 \pm 0.012$ | $0.453 \pm 0.007$ | $0.960 \pm 0.004$ | $0.577 \pm 0.015$ |
| | STEAM $_{\mathrm{ARF}}$ | $0.836 \pm 0.008$ (+0.018) | $0.464 \pm 0.007$ (+0.011) | $0.962 \pm 0.004$ (+0.002) | $0.423 \pm 0.016$ (-0.154) |
| | CTGAN | $0.889 \pm 0.020$ | $0.444 \pm 0.014$ | $0.934 \pm 0.008$ | $0.586 \pm 0.017$ |
| | STEAM $_{\mathrm{CTGAN}}$ | $0.892 \pm 0.017$ (+0.003) | $0.437 \pm 0.012$ (-0.007) | $0.959 \pm 0.005$ (+0.025) | $0.436 \pm 0.012$ (-0.150) |
| | NFlow | $0.817 \pm 0.032$ | $0.418 \pm 0.019$ | $0.913 \pm 0.016$ | $0.643 \pm 0.026$ |
| | STEAM $_{\mathrm{NFlow}}$ | $0.837 \pm 0.040$ (+0.020) | $0.417 \pm 0.015$ (-0.001) | $0.962 \pm 0.005$ (+0.049) | $0.445 \pm 0.020$ (-0.198) |
| | TabDDPM | $0.067 \pm 0.060$ | $0.036 \pm 0.035$ | $0.812 \pm 0.029$ | $1.761 \pm 0.230$ |
| | STEAM $_{\mathrm{TabDDPM}}$ | $0.609 \pm 0.106$ (+0.542) | $0.310 \pm 0.055$ (+0.274) | $0.952 \pm 0.009$ (+0.140) | $0.468 \pm 0.013$ (-1.293) |
| IHDP | CTGAN | $0.663 \pm 0.018$ | $0.419 \pm 0.013$ | $0.888 \pm 0.010$ | $2.521 \pm 0.161$ |
| | STEAM $_{\mathrm{CTGAN}}$ | $0.674 \pm 0.014$ (+0.011) | $0.424 \pm 0.011$ (+0.005) | $0.928 \pm 0.009$ (+0.040) | $1.709 \pm 0.052$ (-0.812) |
| | TabDDPM | $0.477 \pm 0.036$ | $0.340 \pm 0.022$ | $0.862 \pm 0.011$ | $2.706 \pm 0.138$ |
| | STEAM $_{\mathrm{TabDDPM}}$ | $0.553 \pm 0.029$ (+0.076) | $0.396 \pm 0.015$ (+0.056) | $0.918 \pm 0.011$ (+0.056) | $2.346 \pm 0.088$ (-0.360) |
| | ARF | $0.528 \pm 0.009$ | $0.381 \pm 0.010$ | $0.921 \pm 0.009$ | $3.019 \pm 0.117$ |
| | STEAM $_{\mathrm{ARF}}$ | $0.565 \pm 0.014$ (+0.037) | $0.394 \pm 0.010$ (+0.013) | $0.921 \pm 0.009$ (+0.000) | $1.629 \pm 0.056$ (-1.390) |
| | TVAE | $0.622 \pm 0.014$ | $0.410 \pm 0.010$ | $0.880 \pm 0.014$ | $3.198 \pm 0.172$ |
| | STEAM $_{\mathrm{TVAE}}$ | $0.629 \pm 0.015$ (+0.007) | $0.412 \pm 0.011$ (+0.002) | $0.927 \pm 0.007$ (+0.047) | $2.100 \pm 0.075$ (-1.098) |
| | NFlow | $0.406 \pm 0.028$ | $0.309 \pm 0.012$ | $0.882 \pm 0.012$ | $3.835 \pm 0.345$ |
| | STEAM $_{\mathrm{NFlow}}$ | $0.435 \pm 0.034$ (+0.029) | $0.333 \pm 0.020$ (+0.024) | $0.921 \pm 0.007$ (+0.039) | $2.177 \pm 0.118$ (-1.658) |
| ACIC | TVAE | $0.901 \pm 0.014$ | $0.513 \pm 0.004$ | $0.929 \pm 0.005$ | $4.223 \pm 0.138$ |
| | STEAM $_{\mathrm{TVAE}}$ | $0.900 \pm 0.014$ (-0.001) | $0.514 \pm 0.004$ (+0.001) | $0.972 \pm 0.002$ (+0.043) | $2.422 \pm 0.118$ (-1.801) |
| | CTGAN | $0.880 \pm 0.016$ | $0.421 \pm 0.013$ | $0.942 \pm 0.005$ | $4.518 \pm 0.186$ |
| | STEAM $_{\mathrm{CTGAN}}$ | $0.873 \pm 0.014$ (-0.007) | $0.424 \pm 0.014$ (+0.003) | $0.972 \pm 0.002$ (+0.030) | $2.268 \pm 0.154$ (-2.250) |
| | ARF | $0.828 \pm 0.003$ | $0.430 \pm 0.004$ | $0.945 \pm 0.002$ | $4.633 \pm 0.146$ |
| | STEAM $_{\mathrm{ARF}}$ | $0.835 \pm 0.004$ (+0.007) | $0.430 \pm 0.004$ (+0.000) | $0.977 \pm 0.002$ (+0.032) | $2.449 \pm 0.149$ (-2.184) |
| | NFlow | $0.748 \pm 0.019$ | $0.333 \pm 0.014$ | $0.838 \pm 0.035$ | $5.068 \pm 0.282$ |
| | STEAM $_{\mathrm{NFlow}}$ | $0.744 \pm 0.021$ (-0.004) | $0.333 \pm 0.010$ (+0.000) | $0.971 \pm 0.002$ (+0.133) | $2.938 \pm 0.149$ (-2.130) |
| | TabDDPM | $0.124 \pm 0.028$ | $0.002 \pm 0.001$ | $0.813 \pm 0.023$ | $9.281 \pm 1.033$ |
| | STEAM $_{\mathrm{TabDDPM}}$ | $0.141 \pm 0.035$ (+0.017) | $0.002 \pm 0.000$ (+0.000) | $0.955 \pm 0.019$ (+0.142) | $4.497 \pm 0.501$ (-4.784) |

> choice of $Q_{\mathbf{X}}$, as performance significantly differs between different STEAM configurations on the same dataset.

### 7.1.2 Comparison with causal generative models

To fairly compare with CGMs, we must first address their more restrictive assumptions, as discussed in §3. For the ACTG, IHDP, and ACIC data, we do not know the true causal graphs; we simply know which features are the treatment and outcome. To construct reasonable causal graphs using this knowledge, we consider three methods:

1. Construction of a naive graph, $\mathcal{G}_{\mathrm{naive}}$, in which each covariate in $\mathbf{X}$ causes $W$ and $Y$, $W$ causes $Y$, and every pair of covariates has a causal relationship between them;

2. Using the constraint-based PC causal discovery algorithm [89] to discover a graph, $\mathcal{G}_{\mathrm{discovered}}$;

3. Pruning $\mathcal{G}_{\mathrm{discovered}}$ by removing any edges which contradict the DGP we assume, i.e., edges from $Y$ to $W$ or $\mathbf{X}$, or from $W$ to $\mathbf{X}$ are removed, to form $\mathcal{G}_{\mathrm{pruned}}$.

In Table 4, we compare the best STEAM models from Table 3 with two CGMs (using the best of the above graph methods): the additive noise model (ANM) [46] implementation from `DoWhy-GCM` [9], and a diffusion-based causal model (DCM) from [12] (full results in Appendix M).

> **Takeaway.** For each dataset, we see that STEAM outperforms the CGMs in almost every metric, only being outperformed in $P_{\alpha,\mathbf{X}}$ on the ACIC dataset. These results validate that, when the true causal graph is unknown, our less restrictive assumptions enable better generation of synthetic data containing treatments than CGMs.

### 7.2 Comparisons on simulated data

To investigate the performance delta between STEAM and generic generation, we use simulated data with tunable experimental knobs, similar to [14]. These knobs include covariate dimensionality

Table 4: Medical data generation with CGMs and STEAM models, averaged over 20 runs with 95% CIs. Coloured numbers are the relative differences between each CGM and STEAM model.

| Dataset | Model | $P_{\alpha,\mathbf{X}}$ (↑) | $R_{\beta,\mathbf{X}}$ (↑) | $\text{JSD}_\pi$ (↑) | $U_{\text{PEHE}}$ (↓) |
|---|---|---|---|---|---|
| ACTG | STEAM $_{\text{TVAE}}$ | $0.929 \pm 0.008$ | $0.486 \pm 0.009$ | $0.958 \pm 0.004$ | $0.492 \pm 0.011$ |
| | DCM $\mathcal{G}_{\text{pruned}}$ | $0.758 \pm 0.013$ (-0.171) | $0.358 \pm 0.007$ (-0.128) | $0.957 \pm 0.003$ (-0.001) | $0.596 \pm 0.017$ (+0.104) |
| | ANM $\mathcal{G}_{\text{discovered}}$ | $0.836 \pm 0.007$ (-0.093) | $0.419 \pm 0.007$ (-0.067) | $0.952 \pm 0.004$ (-0.006) | $0.578 \pm 0.019$ (+0.086) |
| IHDP | STEAM $_{\text{CTGAN}}$ | $0.674 \pm 0.014$ | $0.424 \pm 0.011$ | $0.928 \pm 0.009$ | $1.709 \pm 0.052$ |
| | DCM $\mathcal{G}_{\text{pruned}}$ | $0.658 \pm 0.011$ (-0.016) | $0.360 \pm 0.007$ (-0.064) | $0.893 \pm 0.008$ (-0.035) | $2.059 \pm 0.140$ (+0.350) |
| | ANM $\mathcal{G}_{\text{pruned}}$ | $0.589 \pm 0.012$ (-0.085) | $0.359 \pm 0.009$ (-0.065) | $0.892 \pm 0.008$ (-0.036) | $1.865 \pm 0.059$ (+0.156) |
| ACIC | STEAM $_{\text{TVAE}}$ | $0.900 \pm 0.014$ | $0.514 \pm 0.004$ | $0.972 \pm 0.002$ | $2.422 \pm 0.118$ |
| | DCM $\mathcal{G}_{\text{discovered}}$ | $0.942 \pm 0.004$ (+0.042) | $0.422 \pm 0.003$ (-0.092) | $0.957 \pm 0.003$ (-0.015) | $4.249 \pm 0.132$ (+1.827) |
| | ANM $\mathcal{G}_{\text{discovered}}$ | $0.929 \pm 0.003$ (+0.029) | $0.404 \pm 0.003$ (-0.110) | $0.872 \pm 0.002$ (-0.100) | $4.193 \pm 0.127$ (+1.771) |

$d$, propensity function $\pi : \mathcal{X}^{(d)} \to [0,1]$, and prognostic and predictive functions $\mu_{\text{prog.}}, \mu_{\text{pred.}} : \mathcal{X}^{(d)} \to \mathbb{R}$.[7] Simulated sample $i$ is generated by drawing $\mathbf{X}^{(i)} \sim \mathcal{N}(0, I_d)$, $W^{(i)} \sim \text{Bern}[\pi(\mathbf{X}^{(i)})]$, and $Y^{(i)} \sim \mathcal{N}(\mu_{\text{prog.}}(\mathbf{X}^{(i)}) + W^{(i)} \cdot \mu_{\text{pred.}}(\mathbf{X}^{(i)}), 1)$. With this DGP, we can assess performance on datasets tailored to specific situations. Across these experiments, we consistently compare between TabDDPM and STEAM$_{\text{TabDDPM}}$, and the default settings for each experimental knob are: $d = 10$, $\pi(\mathbf{X}) = (1 + e^{-1/2(X_1^2 + X_2^2)})^{-1}$, $\mu_{\text{prog.}}(\mathbf{X}) = X_1^2 + X_2^2$, $\mu_{\text{pred.}}(\mathbf{X}) = X_3^2 + X_4^2$.

### 7.2.1 Increasing covariate dimensionality

To investigate performance as $\mathcal{D}_r$ increases in dimensionality, we vary $d \in \{5, 10, 20, 50\}$.

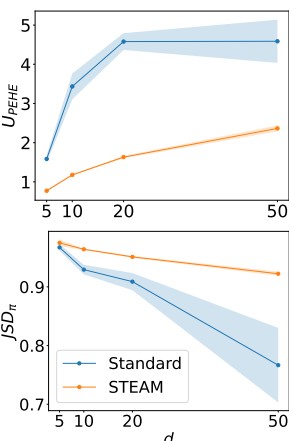

**Takeaway.** The performance delta between STEAM and generic generation grows with the dimensionality of $\mathbf{X}$. This follows the intuition that, as $d$ grows, $P_{\mathbf{X}}$ will dominate the joint distribution, and the comparatively small $P_{W|\mathbf{X}}$ and $P_{Y|W,\mathbf{X}}$ will be overlooked by generic models. The top of Figure 1 shows that, as $d$ increases, both STEAM$_{\text{TabDDPM}}$ and TabDDPM preserve $P_{Y|W,\mathbf{X}}$ worse, but STEAM$_{\text{TabDDPM}}$ is less affected. The bottom of Figure 1 is similar, showing that TabDDPM degrades more than STEAM$_{\text{TabDDPM}}$ in preserving $P_{W|\mathbf{X}}$ as $d$ grows. Direct modelling of these small, but important, distributions by STEAM results in better performance in high dimensions.

Figure 1: $U_{\text{PEHE}}$ and $\text{JSD}_\pi$ as $d$ increases. Averaged over 10 runs, with 95% CIs.

### 7.2.2 Increasing treatment assignment complexity

To investigate performance as $P_{W|\mathbf{X}}$ increases in complexity, we vary the number of covariates upon which it depends. We set $\pi(\mathbf{X}) = (1 + e^{-1/K \sum_{k=1}^{K} X_k^2})^{-1}$ for $K \in \{1, 2, 3, 4, 5\}$.

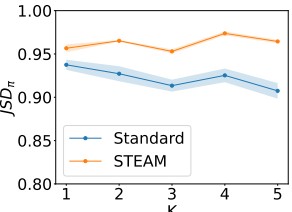

**Takeaway.** STEAM increasingly outperforms generic generation in preserving more complex $P_{W|\mathbf{X}}$. Figure 2 shows that, as $K$ increases, STEAM$_{\text{TabDDPM}}$ maintains a good estimate of $P_{W|\mathbf{X}}$, with $\text{JSD}_\pi$ consistently near 1. TabDDPM degrades with $K$, widening the performance gap. Direct modelling by STEAM allows more complex $P_{W|\mathbf{X}}$ to be preserved.

Figure 2: $\text{JSD}_\pi$ as $K$ increases. Averaged over 10 runs, with 95% CIs.

---

[7]Prognostic variables affect an outcome regardless of treatment, while predictive variables only affect treated outcomes. Prognostic/predictive functions dictate the effects of the relevant variables on the outcome.

### 7.2.3 Outcome heterogeneity

To investigate performance as outcomes become increasingly heterogeneous, we vary the number of covariates upon which $P_{Y|W,\mathbf{X}}$ depends. We set $\mu_{\text{pred.}}(\mathbf{X}) = \sum_{k=3}^{K} X_k^2$ for $K \in \{3, 4, 5, 6, 7\}$.

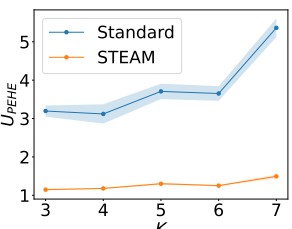

Figure 3: $U_{\text{PEHE}}$ as $K$ increases. Averaged over 10 runs, with 95% CIs.

> **Takeaway.** As $P_{Y|W,\mathbf{X}}$ becomes increasingly heterogeneous, its preservation by STEAM$_{\text{TabDDPM}}$ degrades only slightly, and much more dramatically for TabDDPM, as shown in Figure 3. Again, direct modelling of $Q_{Y|W,\mathbf{X}}$ by STEAM better preserves complex distributions.

The performance delta between STEAM and generic generation grows in complex settings. When generating synthetic copies of real data with high-dimensionality, or complex dependencies in $P_{W|\mathbf{X}}$ or $P_{Y|W,\mathbf{X}}$, STEAM increasingly outperforms. These situations are likely to emerge in real-world data, which is often highly complex, heightening the relevance of the STEAM framework.

## 8 Discussion

In this paper, we tackle a problem impeding progress in the medical community—low-quality synthetic data. Existing methods produce data that are poor for causal inference tasks, which are evaluated with misaligned metrics. Our evaluation metrics and generation framework, grounded in our desiderata which stem from the needs of analysts, remedy this.

We allow meaningful evaluation with our metrics, proposed in §5, that can answer the key questions *Q1-3* of downstream analysts from §1. STEAM generates synthetic data of substantially higher quality than existing methods, demonstrated across a range of experiments in §7, as well as in an ablation study (Appendix J) and hyperparameter stability study (Appendix K). While we focus on medical data, our methods are also applicable to other fields where data contain treatments, or interventions, such as education, marketing, and public policy. In Appendix I we demonstrate performance on the Jobs dataset [61], showing STEAM has similar benefits in non-medical settings.

### 8.1 Future work

There are many future research directions in this setting. In particular, generating synthetic data that respects formal definitions of privacy is an important step. In Appendix N, we prove the immediate compatibility of STEAM with existing differentially private methods, and we conduct initial experiments showing promising results. Developing sophisticated mechanisms for assigning a privacy budget across the component models in STEAM is a potential area of future work, as we discuss in Appendix N.4.

Beyond privacy, extending STEAM to operate in more complicated settings presents several opportunities. One extension, for instance, in settings with a valid instrumental variable $Z$, could be to adapt the STEAM generation process to follow the corresponding causal structure:

$$Z \sim P_Z, \ \mathbf{X} \sim P_{\mathbf{X}|Z}, \ W \sim P_{W|\mathbf{X},Z}, \ Y \sim P_{Y|W,\mathbf{X}}.$$

This would produce synthetic data suitable for treatment effect analysis using instrumental variable methods [38, 30], which do not require the unconfoundedness assumption.

Other extensions could be designed for settings with multiple, or continuous, treatment options. STEAM can easily extend to accommodate multiple treatments by using a multi-class classifier for $Q_{W|\mathbf{X}}$, and PO regressors compatible with $> 2$ treatment arms for $Q_{Y|W,\mathbf{X}}$. More significant architectural changes would be required for continous treatments, as $Q_{W|\mathbf{X}}$ would have to be replaced with a regressor, and $Q_{Y|W,\mathbf{X}}$ would require PO regressors designed for continuous treatments, such as those based on the generalised propensity score [42].

## Acknowledgements

Harry Amad's studentship is funded by Canon Medical Systems Corporation.

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

# A  Example synthetic data papers using medical data containing treatments

Here, we detail selected works in the synthetic data literature that do not consider the downstream task of causal inference. Note that we do not claim that this list is exhaustive, as this is a pervasive problem in the synthetic data literature, and we mean only to provide a few examples here to demonstrate this problem and provide motivation for our paper.

## A.1  MedGAN

In the paper *'Generating Multi-label Discrete Patient Records using Generative Adversarial Networks'* [13], the authors propose a GAN-based approach to generate 'realistic synthetic patient records'. In doing so, they experiment on multiple datasets containing treatments, including one from Sutter Palo Alto Medical Foundation (PAMF), which consists of longitudinal medical records of 258,000 patients, as well as the MIMIC-III dataset [49], which includes 46,000 intensive care unit patient records. Both datasets include treatments administered to patients, and they therefore invite downstream analysts to conduct causal inference tasks, such as treatment effect estimation.

Nevertheless, in this paper, standard generation and evaluation practices are followed, not differentiating between covariates, treatments, and outcomes. In particular, the evaluation protocol relies on marginal comparisons and predictive utility assessment, which offers limited information as to how well the generation method, `medGAN`, produces useful data for causal inference. Furthermore, `medGAN` itself draws samples directly from the joint distribution, not mimicking the DGP of treatment data, or optimising for the distributions most important for causal inference.

## A.2  TabDDPM

The paper *'TabDDPM: Modelling Tabular Data with Diffusion Models'* [58] proposes a diffusion-based tabular data generation method. While not explicitly geared towards medical data, this paper does use medical data with variables that could be seen as treatments in its experiments. It, therefore, at least implicitly, positions itself to work on data containing treatments, and invites users to conduct generation with `TabDDPM` on such data. Specifically, the cardiovascular disease dataset from `https://www.kaggle.com/datasets/sulianova/cardiovascular-disease-dataset` is used, and this data could be analysed via causal inference by setting 'physical activity' as a treatment, to estimate its effect on cardiovascular disease.

However, in this paper, only standard evaluation and generation methods are used, and the needs of downstream analysts pursuing causal inference tasks are not acknowledged. Evaluation involves only predictive utility measures, and the `TabDDPM` method generates all variables in a sample simultaneously, not optimising for the distributions most important for causal inference.

## A.3  GReaT

The paper *'Language Models are Realistic Tabular Data Generators'* [10] proposes an LLM-based generator. Similar to the `TabDDPM` paper, this paper is not explicitly geared towards medical data, but it demonstrates on medical data containing treatments, thereby implicitly condoning its use on this type of data. Specifically, the dataset `sick` from `https://www.openml.org/search?type=data&sort=runs&id=38&status=active` is demonstrated on, which could be analysed via causal inference to assess the effect of 'thyroxine' or 'antithyroid' treatments. Nevertheless, once again, this paper does not consider downstream analysis involving causal inference, only evaluating its `GReaT` method with predictive utility metrics and a discriminator score.

## A.4  Benchmarking process for synthetic electronic health records

Finally, the paper *'A Multifaceted benchmarking of synthetic electronic health record generation models'* proposes a benchmarking framework for use on synthetic electronic health record (EHR) data [98]. Naturally, EHRs will include treatments administered to patients, and they will likely be analysed with treatment effect estimation in mind. In the proposed benchmarking framework, the evaluation procedures—including marginal comparison, correlation comparison, and predictive utility—do not differentiate between covariates, treatments, and outcomes, or acknowledge the needs of downstream analysts conducting causal inference.

# B   Extended literature review

To provide useful context for readers, we extend our literature review here.

**Evaluation.**   We extend on the synthetic data evaluation practices summarised in Table 1 here.

*Marginal comparison.* Assessing the distributional distance between synthetic and real marginals is often used to offer a quantitative assessment of how well individual variables are modelled [98, 91, 32]. The distance function $d$ to conduct this can be set from a variety of choices, including including KL divergence [59], Jensen-Shannon distance [64], Wasserstein distance [53], Kolmogorov-Smirnov score [69], MMD [35], and many more.

*Correlation matrix comparison.* Correlation-based assessment can offer a sense of how well inter-dependencies between variables are modelled in synthetic data [73]. This commonly involves calculating synthetic and real 2-way correlation matrices, and assessing their difference, by setting $d$ as a distance such as Frobenius norm [32] and absolute error [54].

*Joint distribution comparison.* Metrics based on notions of statistical divergence can offer a means of quantifying how different the entire joint distributions of real and synthetic data are [99, 91, 90]. The distance function $d$ can be set to largely the same family of functions as in the marginal comparison case.

*Precision and recall analysis.* Precision and recall, originally proposed for generative model assessment in [82], measure if generated samples are covered by real samples, and vice versa. Alpha precision and beta recall [2] are refined versions of the original metrics, which account for the densities of the real and generative distributions, rather than just comparing supports.

*Discriminator performance.* Discriminator performance is a slightly unique evaluation practice, involving a 'discriminator', which predicts whether instances are synthetic or real, where poor performance of the discriminator indicates realism in the synthetic data [54, 62, 24, 10].

*Predictive utility.* Predictive utility metrics offer a practical evaluation of synthetic data by quantifying the performance of a synthetically-trained predictive model. The "train on synthetic, test on real" (TSTR) paradigm [25] is the common approach to such assessment, measuring the accuracy of a synthetically-trained model in predicting a target label on a real test set.

**Generic generative models.**   We expand on the generic generative modelling paradigm outlined in Table 2 by describing specific existing generative models which adhere to it. These models approximate the real joint distribution using a diverse range of techniques.

*GAN-based models.* GANs [34] consist of a generator and discriminator network, which are trained adversarially to generate and identify synthetic data, respectively, until the samples are realistic. Originally proposed for image generation, GANs have been adapted to tabular generation, and there are many methods that adopt this popular architecture (e.g., CTGAN [97], TableGAN [75]), including those specifically designed for medical data (e.g., MedGAN [13]).

*VAE-based models.* VAEs [56] are another common architecture, which learn to encode data into a lower-dimensional latent space and then decode it back to reconstruct the original data. They generate new data samples by sampling from the latent distribution and decoding these samples, and their application to tabular data involve techniques to handle mixed data types (e.g., TVAE [97]), and regularisation for improved robustness (e.g., RTVAE [1]).

*Diffusion-based models.* Diffusion models [43, 88] learn the gradient of the data distribution, and generate data via progressive denoising, beginning with a noisy sample and using a neural network to predict and remove noise over a number of timesteps, including for tabular data (e.g. TabDDPM [58]). Extensions for direct treatment effect estimation exist [68], but not for synthetic data generation for this purpose.

*Forest-based models.* Random forests can estimate the density of a probability distribution, as leaf nodes partition the data space into distinct hyper-rectangles with estimated densities of the proportion of samples that fall into them. Samples can then be drawn from this estimated density. Random forests can easily handle heterogeneous data types, so their application to tabular data synthesis is natural. They are particularly fast to train and generate from [86, 93]).

*Normalizing flow-based models.* Normalizing flows estimate the target density by transforming a tractable density (e.g., a Gaussian) into the target through a series of invertible transformations, called 'flows'. Probabilities from the target distribution can then be found using the change of variables formula (e.g. [80]). Recent work has shown their theoretical similarity to diffusion models [65].

**Causal generative models.** Causal generative models [57, 9, 12] are a class of generative models, distinct from generic tabular data generators, that approximate the underlying structural causal model (SCM) [76] of a dataset. While such models are related to our work with STEAM and are likely to better preserve causal relationships than generic generators in settings where they can be used, their assumptions can restrict their practical use cases. In comparison to STEAM, they generally differ in terms of their (1) assumptions, (2) motivation, and (3) flexibility, which we detail here.

(1): Importantly, causal generative models typically assume that the data holder has knowledge of the entire causal graph of the real data, which is a more restrictive assumption than we make in this work. We assume that our specification of the underlying DGP $\mathbf{X} \sim P_{\mathbf{X}}$, $W \sim P_{W|\mathbf{X}}$, $Y \sim P_{Y|W,\mathbf{X}}$ is correct for datasets containing treatments, but we do not require knowledge of the causal graph, as we do not need to know the causal links between individual variables. We do not assume knowledge of the causal relationships amongst the covariates, nor knowledge of which covariates cause treatment assignment or patient outcomes. As such, we make less restrictive assumptions than works that require knowledge of a causal graph, and we suggest that our approach is more realistic in complex, real-world scenarios, such as those that arise in medicine, where the true causal graph is unlikely to be available.

(2): The motivation for causal generative models is typically to allow the generation of data to answer graph-specific interventional and counterfactual queries that require knowledge of the full causal graph. With STEAM, however, we seek to generate useful synthetic data only from the observational distribution, for use by analysts with goals such as treatment effect estimation (e.g., CATE).

(3): STEAM's design can, in principle, incorporate any generative model for $Q_{\mathbf{X}}$, essentially acting as a wrapper around $Q_{\mathbf{X}}$ to improve its generation quality for causal inference tasks. This allows STEAM to very easily empower many existing generative modelling frameworks, without having to incorporate bespoke generators. Also, it allows STEAM models to continuously improve along with the base generative model. Existing causal generative models do not generally allow such flexibility, and therefore cannot as easily benefit from future improvements in underlying generic generative models.

Despite these differences, we conduct empirical comparisons between STEAM and some baseline causal generative models in Appendix M.

**Privacy.** Despite the popularity of some of the above generators, memorisation of training samples is a phenomenon observed in generative models [31]. Therefore, provably private generation is often desired to limit the amount of information leaked. Differential privacy [22] is the most common standard adopted, and there are multiple generators which guarantee this, including GAN-based methods (e.g., PATE-GAN [51], DP-GAN [96]) and query-based methods (e.g., GEM [66], MST [70], RAP [5], AIM [71]).

## C  Theorem 1 proof

**Theorem 1.** *Let $P$, $Q^{\theta_1}$, $Q^{\theta_2}$ be of the form described in §5.1, and $\mathcal{M}$ be KL divergence. If we assume that $Q^{\theta_1}$ and $Q^{\theta_2}$ have sufficient capacity to have bounded error on each component, i.e. $\forall i$, $0 < \mathcal{M}(P_{\mathbf{X}_i}, Q_{\mathbf{X}_i}^{\theta \mathbf{X}}) < \varepsilon_{\mathbf{X}}$, and $0 < \mathcal{M}(P_{W|\mathbf{X}}, Q_{W|\mathbf{X}}^{\theta_{W,k}}) < \varepsilon_{W,k}$, and $0 < \mathcal{M}(P_{Y|W,\mathbf{X}}, Q_{Y|W,\mathbf{X}}^{\theta_{Y,k}}) < \varepsilon_{Y,k}$, then:*

$$\frac{\mathcal{M}(P, Q^{\theta_1})}{\mathcal{M}(P, Q^{\theta_2})} \to 1, \ as \ d \to \infty \tag{7}$$

*Proof.* From the factorizations of $P$ and $Q$, KL divergence decomposes:

$$D_{\mathrm{KL}}(P \| Q^{\theta_k}) = \sum_{i=1}^{d} D_{\mathrm{KL}}(P_{\mathbf{X}_i} \| Q_{\mathbf{X}_i}^{\theta \mathbf{X}}) + \mathbb{E}_{P_{\mathbf{X}}}\left[ D_{\mathrm{KL}}(P_{W|\mathbf{x}} \| Q_{W|\mathbf{X}}^{\theta_{W,k}}) \right] + \mathbb{E}_{P_{\mathbf{X},w}}\left[ D_{\mathrm{KL}}(P_{Y|W,\mathbf{x}} \| Q_{Y|W,\mathbf{X}}^{\theta_{Y,k}}) \right] \tag{8}$$

As such, the following holds for when KL divergence is set as $\mathcal{M}$.

Define the ratio:

$$R(d) = \frac{\mathcal{M}(P, Q^{\theta_1})}{\mathcal{M}(P, Q^{\theta_2})}.$$

Substituting the decompositions, we have:

$$R(d) = \frac{\sum_{i=1}^{d} \mathcal{M}(P_{\mathbf{X}_i}, Q_{\mathbf{X}_i}^{\theta \mathbf{X}}) + \mathbb{E}_{P_X}\left[ \mathcal{M}(P_{W|\mathbf{X}}, Q_{W|\mathbf{X}}^{\theta_{W,1}}) \right] + \mathbb{E}_{P_{X,w}}\left[ \mathcal{M}(P_{Y|W,\mathbf{X}}, Q_{Y|W,\mathbf{X}}^{\theta_{Y,1}}) \right]}{\sum_{i=1}^{d} \mathcal{M}(P_{\mathbf{X}_i}, Q_{\mathbf{X}_i}^{\theta \mathbf{X}}) + \mathbb{E}_{P_X}\left[ \mathcal{M}(P_{W|\mathbf{X}}, Q_{W|\mathbf{X}}^{\theta_{W,2}}) \right] + \mathbb{E}_{P_{X,w}}\left[ \mathcal{M}(P_{Y|W,\mathbf{X}}, Q_{Y|W,\mathbf{X}}^{\theta_{Y,2}}) \right]}.$$

As the dimensionality $d$ increases, the marginal summations $\sum_{i=1}^{d} \mathcal{M}(P_{\mathbf{X}_i} \| Q_{\mathbf{X}_i}^{\theta \mathbf{X}})$ grow linearly with $d$, since each $\mathcal{M}(P_{\mathbf{X}_i} \| Q_{\mathbf{X}_i}^{\theta \mathbf{X}})$ is, by assumption, non-negative, and they therefore dominate the bounded conditional contributions:

$$\mathbb{E}_{P_X}\left[ \mathcal{M}(P_{W|\mathbf{X}}, Q_{W|\mathbf{X}}^{\theta_{W,k}}) \right] < \varepsilon_{W,k},$$

$$\mathbb{E}_{P_{X,w}}\left[ \mathcal{M}(P_{Y|W,\mathbf{X}}, Q_{Y|W,\mathbf{X}}^{\theta_{Y,k}}) \right] < \varepsilon_{Y,k}.$$

Thus, $\mathcal{M}(P_{\mathbf{X},W,Y}, Q_{\mathbf{X},W,Y}^{\theta_k}) \sim \sum_{i=1}^{d} \mathcal{M}(P_{\mathbf{X}_i}, Q_{\mathbf{X}_i}^{\theta \mathbf{X}})$ and $R(d) \sim \frac{\sum_{i=1}^{d} \mathcal{M}(P_{\mathbf{X}_i}, Q_{\mathbf{X}_i}^{\theta \mathbf{X}})}{\sum_{i=1}^{d} \mathcal{M}(P_{\mathbf{X}_i}, Q_{\mathbf{X}_i}^{\theta \mathbf{X}})} = 1.$

Therefore:

$$R(d) \to 1, \ as \ d \to \infty$$

$\square$

## D   Empirical demonstrations of current metric failure

### D.1   Failure to identify changes to the outcome generation mechanism

We demonstrate this with a simple experiment investigating how four $\mathcal{D}_s$ of size $n = 1000$, which only differ in their outcome generation mechanisms, are assessed by an array of current metrics. We simulate $\mathcal{D}_r$ from a simple DGP with 10 covariates with $P_{\mathbf{X}} = \mathcal{N}(0, I)$, $P_{W|\mathbf{X}} = \text{Bern}(0.5)$, $P_{Y|W,\mathbf{X}} = \mathcal{N}(W \cdot X_1{}^2, 1)$. We generate four $\mathcal{D}_s^i$ with the same $Q_{\mathbf{X}}^i \overset{d}{=} P_{\mathbf{X}}$, $Q_{W|\mathbf{X}}^i \overset{d}{=} P_{W|\mathbf{X}}$, $\forall i \in \{1, 2, 3, 4\}$. We vary each $Q_{Y|W,\mathbf{X}}^i \sim \mathcal{N}(W \cdot \Phi_i(\mathbf{X}, 1) + (1 - W) \cdot \Phi_i(\mathbf{X}, 0), 1)$ where $\Phi_i$ represents a potential outcome (PO) estimator with the architecture from either an S-Learner, T-Learner [60], DragonNet [85], or TARNet [84]. These four architectures will model $Q_{Y|W,X}^i$ differently, inducing the only point of variation amongst the $\mathcal{D}_s^i$.

Since we simulate $\mathcal{D}_r$, we know the ground-truth treatment effects, and an oracle metric can be established to determine the true quality of each $\mathcal{D}_s^i$. We define this as the precision of estimating heterogeneous effects (PEHE) [41] of estimates from a CATE learner trained on $\mathcal{D}_s^i$ and the ground-truth CATEs. In Table 5 we report the scores of $P_\alpha$, $R_\beta$ [2], inverse KL divergence [59], Kolmogorov-Smirnov (KS) score [69], Wasserstein distance (WD) [53], and Jensen-Shannon distance (JSD) [64] on each $\mathcal{D}_s^i$. All report very similar scores across the $\mathcal{D}_s^i$, with most offering no statistically significant best option, suggesting that their quality is the same. The oracle metric, however, determines that $\mathcal{D}_s^i$ using a T-Learner for $\Phi_i$ is a clear best, and $\mathcal{D}_s^i$ with an S-Learner as $\Phi_i$ is more than twice as bad at preserving the true treatment effects. Clearly, even in a moderately sized dataset, these metrics cannot reliably identify changes in $Q_{Y|W,\mathbf{X}}^i$, despite the large effect that this distribution has on downstream performance.

Table 5: Joint-distribution-level metrics on $\mathcal{D}_s^i$ which differ in $Q_{Y|W,\mathbf{X}}^i$ architecture only. Averaged over 10 runs, with 95% CIs.

| $Q_{Y|W,\mathbf{X}}^i$ | $P_\alpha$ ($\uparrow$) | $R_\beta$ ($\uparrow$) | Inv. KL ($\uparrow$) | KS ($\uparrow$) | WD ($\downarrow$) | JSD ($\downarrow$) | Oracle ($\downarrow$) |
|---|---|---|---|---|---|---|---|
| T-Learner | $0.927 \pm 0.001$ | $0.584 \pm 0.006$ | $0.947 \pm 0.000$ | $0.979 \pm 0.000$ | $0.002 \pm 0.000$ | $0.002 \pm 0.000$ | $0.525 \pm 0.012$ |
| TARNet | $0.919 \pm 0.002$ | $0.573 \pm 0.005$ | $0.950 \pm 0.006$ | $0.985 \pm 0.001$ | $0.002 \pm 0.000$ | $0.002 \pm 0.000$ | $0.616 \pm 0.015$ |
| DragonNet | $0.921 \pm 0.001$ | $0.574 \pm 0.004$ | $0.947 \pm 0.000$ | $0.984 \pm 0.001$ | $0.002 \pm 0.000$ | $0.002 \pm 0.000$ | $0.618 \pm 0.007$ |
| S-Learner | $0.926 \pm 0.002$ | $0.579 \pm 0.007$ | $0.957 \pm 0.009$ | $0.990 \pm 0.000$ | $0.002 \pm 0.000$ | $0.001 \pm 0.000$ | $1.279 \pm 0.015$ |

**Comparison to $U_{\text{PEHE}}$.**   Since we only alter $Q_{Y|W,\mathbf{X}}^i$ between each $\mathcal{D}_s^i$, all have the same $P_{\alpha,\mathbf{X}}$, $R_{\beta,\mathbf{X}}$, and $\text{JSD}_\pi$. In Table 6 we report $U_{\text{PEHE}}$ on each dataset, and we see that it fully reproduces the oracle ranking, and correctly identifies the best dataset to a statistically significant level, which no existing metric could do.

Table 6: $U_{\text{PEHE}}$ on $\mathcal{D}_s^i$ with varied $Q_{Y|W,\mathbf{X}}^i$. Averaged over 10 runs, with 95% CIs.

| $Q_{Y|W,\mathbf{X}}^i$ | $U_{\text{PEHE}}$ ($\downarrow$) | Oracle ($\downarrow$) |
|---|---|---|
| T-Learner | $0.693 \pm 0.013$ | $0.525 \pm 0.012$ |
| TARNet | $0.731 \pm 0.016$ | $0.616 \pm 0.015$ |
| DragonNet | $0.754 \pm 0.019$ | $0.618 \pm 0.007$ |
| S-Learner | $0.906 \pm 0.019$ | $1.279 \pm 0.015$ |

### D.2   Failure to identify changes to the treatment assignment mechanism

We conduct a similar experiment varying $Q_{W|\mathbf{X}}^i$ across three $\mathcal{D}_s^i$. We simulate $\mathcal{D}_r \sim P_{\mathbf{X},W,Y}$ from a DGP with 5 covariates, all of which contribute to the propensity score. We set $P_{\mathbf{X}} = \mathcal{N}(0, I)$, $P_{W|\mathbf{X}} = \text{Bern}(\pi(\mathbf{X}))$, $\pi(\mathbf{X}) = (1 + e^{-1/5 \sum_{i=1}^{5} X_i})^{-1}$, $P_{Y|W,\mathbf{X}} = \mathcal{N}(0, 1)$. We generate three $\mathcal{D}_s^i \sim Q_{\mathbf{X},W,Y}^i$ which vary only in the degree to which they correctly model $\pi(\mathbf{X})$ by setting $Q_{\mathbf{X}}^i \overset{d}{=} P_{\mathbf{X}}$, $Q_{Y|W,\mathbf{X}}^i \overset{d}{=} P_{Y|W,\mathbf{X}}$, $\forall i \in \{1, 2, 3\}$ and $Q_{W|\mathbf{X}}^i = \text{Bern}(\pi_i(\mathbf{X}))$ where $\pi_1(\mathbf{X}) = (1 + e^{-X_1})^{-1}$, $\pi_2(\mathbf{X}) = (1 + e^{-1/3 \sum_{i=1}^{3} X_i})^{-1}$, and $\pi_3(\mathbf{X}) = (1 + e^{-1/5 \sum_{i=1}^{5} X_i})^{-1}$. In this way, we know that,

Table 7: # correct var.: The number of correctly identified variables in the propensity score. $P_\alpha$: $\alpha$ precision. $R_\beta$: $\beta$ recall. Inv. KL: Inverse KL divergence. KS: Kolmogorov-Smirnov score. WD: Wasserstein distance. JSD: Jensen-Shannon distance. $\text{JSD}_\pi$: Ours. Averaged over 10 runs, with 95% CIs.

| # correct var. | $P_\alpha$ ($\uparrow$) | $R_\beta$ ($\uparrow$) | Inv. KL ($\uparrow$) | KS ($\uparrow$) | WD ($\downarrow$) | JSD ($\downarrow$) | $\text{JSD}_\pi$ ($\uparrow$) |
|---|---|---|---|---|---|---|---|
| 5 | $0.863 \pm 0.024$ | $0.456 \pm 0.017$ | $0.989 \pm 0.002$ | $0.965 \pm 0.003$ | $0.017 \pm 0.001$ | $0.004 \pm 0.000$ | $0.963 \pm 0.007$ |
| 3 | $0.868 \pm 0.022$ | $0.457 \pm 0.012$ | $0.981 \pm 0.006$ | $0.966 \pm 0.003$ | $0.018 \pm 0.001$ | $0.004 \pm 0.000$ | $0.942 \pm 0.006$ |
| 1 | $0.866 \pm 0.021$ | $0.453 \pm 0.013$ | $0.985 \pm 0.003$ | $0.965 \pm 0.003$ | $0.018 \pm 0.001$ | $0.004 \pm 0.000$ | $0.908 \pm 0.013$ |

in truth, $Q^3_{\mathbf{X},W,Y}$ is a better model than $Q^2_{\mathbf{X},W,Y}$, which in turn is better than $Q^1_{\mathbf{X},W,Y}$, and we can now assess how well existing metrics, and our $\text{JSD}_\pi$ metric, recover this ranking.

We display the scores of $P_\alpha$, $R_\beta$, inverse KL, Kolmogorov-Smirnov score, Wasserstein distance, Jensen-Shannon distance, and our metric $\text{JSD}_\pi$ on each $\mathcal{D}^i_s$ in Table 7. We see that the existing metrics report very similar scores across the three datasets, and none offer a statistically significant best option. This contrasts with our $\text{JSD}_\pi$ metric, which correctly orders the three models and selects $Q^3_{\mathbf{X},W,Y}$ as the best option to a statistically significant level.

To further elucidate the differences in rankings between existing and our metrics, both in this experiment and the outcome generation comparison in §5, we list each ranking and their Spearman's rank correlation coefficient with the oracle ranking in Tables 8 and 9. Assessment via our metrics is the only protocol that reproduces the oracle ranking across both experiments.

Table 8: Treatment assignment experiment: rankings by different metrics, sorted by Spearman's rank correlation coefficient ($r_s$) with oracle ranking. Numbering indicates the oracle order of $\pi_i(\mathbf{X})$.

| Metric | Ranking | $r_s$ ($\uparrow$) |
|---|---|---|
| $P_\alpha$ | 2,3,1 | -0.5 |
| KS | 2,1,3 | 0.5 |
| Inv. KL | 1,3,2 | 0.5 |
| $R_\beta$ | 2,1,3 | 0.5 |
| WD | 1,2,3 | 1 |
| $\text{JSD}_\pi$ | 1,2,3 | 1 |

Table 9: Outcome generation experiment: Rankings by different metrics, sorted by Spearman's rank correlation coefficient ($r_s$) with oracle ranking. $Q^i_{Y|W,\mathbf{X}}$ are numbers by oracle ranking, 1: T-Learner, 2: TARNet, 3: DragonNet, 4: S-Learner.

| Metric | Ranking | $r_s$ ($\uparrow$) |
|---|---|---|
| KS | 4,2,3,1 | -0.80 |
| Inv. KL | 4,2,1,3 | -0.40 |
| $P_\alpha$ | 1,4,3,2 | 0.20 |
| $R_\beta$ | 1,4,3,2 | 0.20 |
| WD | 2,3,1,4 | 0.40 |
| $U_{\text{PEHE}}$ | 1,2,3,4 | 1 |

### D.3   Existing metric failure: extreme example

As a 'proof by contradiction' that current metrics can offer a good level of information on the preservation of (i)–(iii), we present some extreme examples. We show that joint-distribution-level metrics do not have enough resolution to identify how well (i)–(iii) are preserved, even if $Q$ comprehensively fails in modelling any one of the component distributions $P_{\mathbf{X}}$, $P_{W|\mathbf{X}}$, or $P_{Y|W,\mathbf{X}}$.

We perform a series of experiments where we evaluate adversarial synthetic versions of a simulated dataset, with each synthetic version failing in one of the above components, and we show that standard metrics do not identify these failure modes. We simulate real data using the DGP in CATENets

from [15] and we create three $\mathcal{D}_s$ that perfectly model two component distributions of $\mathcal{D}_r$ but poorly approximate the remaining one. For poorly modelled $P_{\mathbf{X}}$, we set $\mathbf{X} = \mathbf{0}$; for poorly modelled $P_{W|\mathbf{X}}$, we assign all instances with $W = 0$; and for poorly modelled $P_{Y|W,\mathbf{X}}$, we draw $Y$ from a normal distribution with mean 0 regardless of treatment. All such $\mathcal{D}_s$ are useless for treatment effect estimation.

Table 10: Scores on adversarially created $\mathcal{D}_s$ which poorly perform on desiderata (i), (ii), or (iii).

|            | **Inv. KL** ($\uparrow$) | $\boldsymbol{P_\alpha}$ ($\uparrow$) | $\boldsymbol{R_\beta}$ ($\uparrow$) | **MMD** ($\downarrow$) |
|------------|------|------|------|------|
| Poor (i)   | 0.681 | 0.902 | 0.368 | 0.085 |
| Poor (ii)  | 0.685 | 0.501 | 0.333 | 0.074 |
| Poor (iii) | 0.844 | 0.905 | 0.430 | 0.008 |

We report the inverse of KL divergence, $P_\alpha$, $R_\beta$, and MMD, which all have range $[0, 1]$, for these synthetic datasets in Table 10. We see that these conventional evaluation metrics do not accurately reflect the invalidity of each $\mathcal{D}_s$ for treatment effect estimation. None report significantly low scores, despite the failure of each $\mathcal{D}_s$. $R_\beta$ reflects these failures best, although its scores still do not adequately reflect how these datasets render correct treatment effect analysis impossible, and it does not allow a granular enough analysis to disentangle which component distribution is poorly modelled.

In further detail, for the experiments that examine poor modelling of $P_{\mathbf{X}}$ and $P_{W|\mathbf{X}}$, we simulate $\mathcal{D}_r$ of size $n = 1000$ with $d = 1$ covariate as follows:

$$X \sim \mathcal{N}(0, 1) \tag{9}$$
$$W \sim \text{Bernoulli}(0.5) \tag{10}$$
$$Y(0), Y(1) = 0 \tag{11}$$
$$Y = (1 - W)Y(0) + WY(1) + \epsilon, \ \epsilon \sim \mathcal{N}(0, 1) \tag{12}$$

We manufacture $\mathcal{D}_s$ that exhibits poor modelling of $P_{\mathbf{X}}$ by generating $W$ and $Y$ from the true distributions as above, but set $\mathbf{X} = \mathbf{0}$. For $\mathcal{D}_s$ that exhibits poor modelling of $P_{W|\mathbf{X}}$, we generate $\mathbf{X}$ and $Y$ from their true distributions, but set all $W = 0$.

To demonstrate assessment under poor modelling of $P_{Y|W,\mathbf{X}}$, we set the covariate in $\mathcal{D}_r$ to be predictive, such that it affects the value of the potential outcome $Y(1)$, but not $Y(0)$. The distributions remain the same as the above, although now the potential outcomes are:

$$Y(0) = 0 \tag{13}$$
$$Y(1) = X^2 \tag{14}$$

We manufacture $\mathcal{D}_s$ that poorly models of $P_{Y|W,\mathbf{X}}$ by generating $\mathbf{X}$ and $W$ from their true distributions, but we set $Y(0), Y(1) = 0$.

# E  Discussion on alternative metrics

While we propose a set of metrics $\mathcal{M}$ for evaluation of $\mathcal{D}_\text{s}$, there are many possible alternatives to each choice we make. Our choices enable evaluation of how well $\mathcal{D}_\text{s}$ adheres to our desiderata, but, like any metrics, they may be sub-optimal for certain data holders with specific preferences. Here, we list some alternative definitions, and we detail when they may be preferable. We would like to emphasise that conducting *any* reasonable assessment of the preservation of (i)–(iii) is beneficial compared to standard evaluation practices.

## E.1  Alternative covariate distribution assessment

As we state in the main paper, comparison of $P_X$ and $Q_X$ is essentially a standard synthetic data evaluation problem, and therefore any standard protocol can be applied.

For example, if the dimensionality of $\mathbf{X}$ is small, manual evaluation via visualisation may be preferable to the precision/recall analysis we suggest, as this can provide a more granular and interpretable assessment. On the other hand, if a single all-encompassing score is desired, rather than the two-dimensional metric $(P_{\alpha,\mathbf{X}}, R_{\beta,\mathbf{X}})$, then statistical divergence metrics can offer this. These one-dimensional metrics can lead to more straightforward model selection than $(P_{\alpha,\mathbf{X}}, R_{\beta,\mathbf{X}})$, as ordering based on a two-dimensional metric can be ambiguous.

## E.2  Alternative treatment assignment mechanism assessment

Similarly, there is a vast array of metrics which could be substituted into Eq. (5) over Jensen-Shannon distance, which could measure the difference between $P_{W|\mathbf{X}}$ and $Q_{W|\mathbf{X}}$. These include metrics such as KL divergence and Wasserstein distance, which are also very common in machine learning literature. For example, a data holder may prefer KL divergence if they want to more harshly punish $Q_{W|\mathbf{X}}$ for failing to place density where $P_{W|\mathbf{X}}$ is probable, encouraging *mode-covering* behaviour. On the other hand, if a data holder wants to more harshly punish $Q_{W|\mathbf{X}}$ for spreading mass away from the modes, Wasserstein distance may be preferable, leading to *mode-seeking* behaviour. JSD achieves a balance between these two focuses, but if a data holder has a strong preference for one or the other, these alternate choices would be preferable. Nevertheless, we suggest that, apart from extreme scenarios, most reasonable methods to assess the preservation of $P_{W|\mathbf{X}}$ will lead to similar analysis.

## E.3  Alternative outcome generation mechanism metrics

Raw similarity of PEHE in CATE estimation between $\mathcal{D}_\text{r}$ and $\mathcal{D}_\text{s}$ may not be the most important quantity of interest for certain data holders. This can be particularly true in medical practice, as raw performance is not the only important aspect of a downstream model. We propose some alternatives that may be more applicable in the following situations:

1. Correct estimation of the *sign* of the CATE may be of heightened importance if the CATE learner is assisting with policy decisions. The wrong CATE sign will lead to incorrect policy administration, whereas the magnitude of the effect may not be as important for decision-making.

2. Discovering the correct drivers of effect heterogeneity may be important, as how a learner arrives at its final estimation is particularly important to consider in applications such as in drug discovery or clinical practice [40, 14].

### E.3.1  Policy assignment

If policy guidance is of interest, then quantification of how well the sign of CATE estimates is preserved between $\mathcal{D}_\text{s}$ and $\mathcal{D}_\text{r}$ may be desired, which can be done as follows:

$$U_{\text{policy}}(\mathcal{D}_\text{r}, \mathcal{D}_\text{s}) = \frac{1}{|\mathcal{F}|} \sum_{\hat{\tau} \in \mathcal{F}} \mathbb{E}_{P_\mathbf{X}} [I(\hat{\tau}_{\text{synth}}(\mathbf{X}) \times \hat{\tau}_{\text{real}}(\mathbf{X}) > 0)] \tag{15}$$

where $I$ is the indicator function.

### E.3.2 Feature importance

If assessing how well $\mathcal{D}_s$ permits the discovery of the correct drivers of effect heterogeneity is important, this can be quantified through the use of feature importance methods. Given a CATE learner $\hat{\tau}$, feature importance methods offer a means to measure the sensitivity of the model to each covariate by assigning an importance score $a_i(\hat{\tau}, \mathbf{x})$ to each feature $x_i$ that reflects its importance in the prediction of the CATE $\hat{\tau}(\mathbf{x})$. There are many different instantiations of feature importance methods with different strengths [26], and the metric we propose here is method-agnostic. We quantify how well $P_{Y|W,\mathbf{X}}$ is modelled according to feature importance similarity between $\mathcal{D}_s$ and $\mathcal{D}_r$ as follows:

$$U_{\text{int}}(\mathcal{D}_r, \mathcal{D}_s) = \frac{1}{|\mathcal{F}|} \sum_{\hat{\tau} \in \mathcal{F}} S_C(A_{\text{real}, \hat{\tau}}, A_{\text{synth}, \hat{\tau}}) \tag{16}$$

where $S_C$ is cosine similarity, and $A_{\text{real}, \hat{\tau}}$ and $A_{\text{synth}, \hat{\tau}}$ are $d$-dimensional vectors with $i^{th}$ entries

$$A^i_{\diamond, \hat{\tau}} = \mathbb{E}_{P_{\mathbf{X}}}[a_i(\hat{\tau}_\diamond, \mathbf{X})], \diamond \in \{\text{real}, \text{synth}\} \tag{17}$$

$U_{\text{int}} \in [-1, 1]$, where $U_{\text{int}} = 1$ indicates total agreement in the feature importances of $\mathcal{D}_r$ and $\mathcal{D}_s$, while $U_{\text{int}} = 0$ indicates that the feature importances are uncorrelated, suggesting that $Q_{Y|W,\mathbf{X}}$ does not allow discovery of the correct drivers of heterogeneity. Finally, $U_{\text{int}} = -1$ indicates antithetical feature importances, suggesting a drastic failure of $Q_{Y|W,\mathbf{X}}$ in estimating $P_{Y|W,\mathbf{X}}$.

# F  Defining $\mathcal{F}$ for $U_{\text{PEHE}}$

In CATE estimation, model validation is a difficult task [16]. As such, it is reasonable to expect that a set of downstream analysts conducting CATE estimation on $\mathcal{D}_\text{s}$ will use different learners. Therefore, we want $U_{\text{PEHE}}$ to reflect the expected difference in downstream performance between $\mathcal{D}_\text{s}$ and $\mathcal{D}_\text{r}$ across a diverse array of potential learners, such that it is representative for the entire population of analysts, and has limited bias towards any particular learner class. To achieve this, we propose averaging $U_{\text{PEHE}}$ across a family of CATE learners $\mathcal{F}$, and we suggest that larger $|\mathcal{F}|$ and a diverse selection of the learners within $\mathcal{F}$ is preferable.

Of course, there is a trade-off between the size of $\mathcal{F}$, and therefore the stability of $U_{\text{PEHE}}$, and the computational cost of repeated CATE estimation. With this in mind, to limit the computation involved in calculating $U_{\text{PEHE}}$, we suggest that users should be selective of the learners included in $\mathcal{F}$ to maximize learner diversity, and minimise $|\mathcal{F}|$. For example, in our experiments, we set $|\mathcal{F}| = 4$, and we chose learners from both of the high-level CATE learning strategies described in [15] (i.e., one-step plug-in learners, and two-step learners). Specifically, for the one-step learners we use S- and T-learners [60], and for the two-step learners we use RA- and DR-learners [55]. All four of these learners conduct CATE estimation differently, and encode different inductive biases in their approaches, and thus they form a good diverse base for $\mathcal{F}$.

For our experiments, on each of the real datasets, the runtime for calculating $U_{\text{PEHE}}$ for one run is shown in Table 11. Note that these are much less than the typical generation times for each dataset, so this step is unlikely to be a large time burden for the data holder. Also note that these calculations can be parallelised across the learner classes, which we did not do, and this can improve the computational feasibility of using a larger $|\mathcal{F}|$.

Table 11: Runtime to calculate $U_{\text{PEHE}}$

| Dataset | $U_{\text{PEHE}}$ runtime (s) |
|---------|------------------------------|
| ACTG    | 26  |
| IHDP    | 60  |
| ACIC    | 191 |

# G    STEAM diagram

See Figure 4 for flowcharts of the generic DGP for synthetic data generation, real datasets containing treatments, and STEAM. STEAM is designed to closely mimic the real DGP.

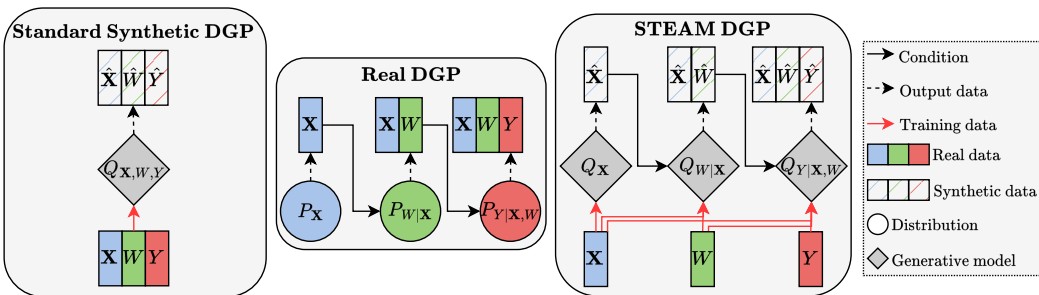

Figure 4: DGPs for generic generative models (left), real datasets (middle), and STEAM (right).

# H Main experimental details

Here we add any additional details to the experiment set-ups from §7. All experiments were run on an Azure VM with a 48-Core AMD EPYC Milan CPU and an A100 GPU. We report typical runtimes where relevant. An estimated total compute time for all experimental runs is ∼72 hours. This does not include the compute required for preliminary experimentation.

For all generative models, we use the open source library `synthcity` [79] (Apache-2.0 License), and we do not change the default hyperparameters. We set the treatment and outcome generators of STEAM using a logistic regression function from `scikit-learn` [77] and S-Learner from `CATENets` [15], respectively.

We release code here `https://github.com/harrya32/STEAM` and here `https://github.com/vanderschaarlab/STEAM`.

## H.1 Generation of medical data containing treatments

To assess sequential generation in a number of real-world scenarios, we evaluate performance on ACTG [36] and on the popular treatment effect estimation datasets IHDP [41] and ACIC [21]. We also report further results in Table 12 on a non-medical dataset, Jobs [61], which is also popular amongst the treatment effect estimation community, to show that STEAM can be applied beyond the medical context, to any dataset containing treatments. More in-depth descriptions of the datasets used are here:

1. **AIDS Clinical Trial Group (ACTG) study 175.** A clinical trial on subjects with HIV-1 [36]. Preprocessed as in [39] to compare CD4 counts at the beginning of the study and after $20 \pm 5$ weeks across treatment arms using zidovudine (ZDV) and zalcitabine (ZAL) vs. ZDV only. The ACTG dataset contains $n = 1056$ instances with $d = 12$ covariates and a continuous outcome, and we use the publicly available version from `https://github.com/tobhatt/CorNet`.

2. **Infant Health and Development Program (IHDP).** A semi-synthetic medical dataset, with real covariates and simulated outcomes, using data from a randomised experiment designed to evaluate the effect of specialist childcare on the cognitive test scores of premature infants [11]. Confounding and treatment imbalance were introduced in [41] to mimic an observational dataset. The IHDP dataset consists of $n = 747$ instances with $d = 25$ covariates and a continuous outcome. We use the publicly available version from `https://github.com/AMLab-Amsterdam/CEVAE` [67], with the first batch of simulated outcomes.

3. **Atlantic Causal Inference Competition 2016 (ACIC).** A semi-synthetic medical dataset, with real covariates and simulated outcomes, containing data from the Collaborative Perinatal Project [74]. The data was modified in [21] to simulate an observational study examining the impact of birth weight in twins on IQ. The ACIC dataset consists of $n = 4802$ instances with $d = 58$ covariates and a continuous outcome. We use the publicly available version from the `causallib` package [87] (Apache-2.0 License) available here `https://github.com/BiomedSciAI/causallib`, using the first simulated set of treatments and potential outcomes.

4. **Jobs.** Jobs contains experimental data from a male sub-sample from the National Supported Work Demonstration from [61] to evaluate the effect of job training on income. The Jobs dataset consists of $n = 722$ instances with $d = 7$ covariates and a continuous outcome. We use the publicly available version used in [18, 19], from `https://users.nber.org/~rdehejia/data/.nswdata2.html`.

We report extended results for all models tested, and further results on the Jobs dataset, in Table 12. For each model on each dataset, we conduct 20 runs. A typical run for a given real dataset and generative model took 15 minutes.

## H.2 Simulated experiments

For our simulated insight experiments, we compare the performance of a standard TabDDPM with STEAM$_{\text{TabDDPM}}$, and we report average results over 10 runs. A typical run took 5 minutes. For simulation of $\mathcal{D}_{\text{r}}$, we use the DGP from `CATENets` [15].

## H.3 Differentially private generation

For our experiment, which showcases the performance of STEAM when satisfying DP (Appendix N), we compare the generative performance of baseline methods AIM [71], GEM [66], MST [70], RAP [5] with their STEAM counterparts. We use the code provided by [71] in their GitHub `https://github.com/ryan112358/private-pgm` for the AIM and MST implementations, and we use the code provided in the GitHub `https://github.com/terranceliu/dp-query-release` for the GEM and RAP implementations. We use the default hyperparameter settings of these implementations, with the workload set as 3-way marginals. For the STEAM models, we use the relevant base model for $Q_{\mathbf{X}}$, DP random forest from the `diffprivlib` library [45] (MIT License) for $Q_{W|\mathbf{X}}$, and a custom implementation of a T-Learner [60] based on [15] which guarantees DP by training with DP stochastic gradient descent, implemented with the `Opacus` library [100] (Apache-2.0 License). We report comparative results on varied $\epsilon$, averaged over 5 runs.

# I    Extended results

We report the full set of results for each model and dataset from §7.1 in Table 12, including on the Jobs dataset. We pair each standard model with its STEAM analogue, and report the relative difference between them for each metric, where (green) indicates better performance by STEAM, and (red) indicates better performance by standard modelling. We see that STEAM clearly outperforms. Almost all STEAM models perform better in each metric than all standard models.

Table 12: $P_{\alpha,\mathbf{X}}$, $R_{\beta,\mathbf{X}}$, $\text{JSD}_\pi$, and $U_{\text{PEHE}}$ values for STEAM and standard models. Averaged over 20 runs, with 95% confidence intervals. Each STEAM model is placed after its corresponding standard model. Coloured numbers in brackets indicate relative difference between standard and STEAM model, where (green) indicates better performance by STEAM, and (red) indicates better performance by standard modelling.

| Dataset | Model | $P_{\alpha,\mathbf{X}}$ (↑) | $R_{\beta,\mathbf{X}}$ (↑) | $\text{JSD}_\pi$ (↑) | $U_{\text{PEHE}}$ (↓) |
|---|---|---|---|---|---|
| ACTG | TVAE | $0.926 \pm 0.013$ | $0.483 \pm 0.010$ | $0.946 \pm 0.004$ | $0.564 \pm 0.017$ |
| | STEAM $_{\text{TVAE}}$ | $0.929 \pm 0.008$ (+0.003) | $0.486 \pm 0.009$ (+0.003) | $0.958 \pm 0.004$ (+0.012) | $0.492 \pm 0.011$ (-0.072) |
| | ARF | $0.818 \pm 0.012$ | $0.453 \pm 0.007$ | $0.960 \pm 0.004$ | $0.577 \pm 0.015$ |
| | STEAM $_{\text{ARF}}$ | $0.836 \pm 0.008$ (+0.018) | $0.464 \pm 0.007$ (+0.011) | $0.962 \pm 0.004$ (+0.002) | $0.423 \pm 0.016$ (-0.154) |
| | CTGAN | $0.889 \pm 0.020$ | $0.444 \pm 0.014$ | $0.934 \pm 0.008$ | $0.586 \pm 0.017$ |
| | STEAM $_{\text{CTGAN}}$ | $0.892 \pm 0.017$ (+0.003) | $0.437 \pm 0.012$ (-0.007) | $0.959 \pm 0.005$ (+0.025) | $0.436 \pm 0.012$ (-0.150) |
| | NFlow | $0.817 \pm 0.032$ | $0.418 \pm 0.008$ | $0.913 \pm 0.016$ | $0.643 \pm 0.026$ |
| | STEAM $_{\text{NFlow}}$ | $0.837 \pm 0.040$ (+0.020) | $0.417 \pm 0.015$ (-0.001) | $0.962 \pm 0.005$ (+0.049) | $0.445 \pm 0.020$ (-0.198) |
| | TabDDPM | $0.067 \pm 0.060$ | $0.036 \pm 0.035$ | $0.812 \pm 0.029$ | $1.761 \pm 0.230$ |
| | STEAM $_{\text{TabDDPM}}$ | $0.609 \pm 0.106$ (+0.542) | $0.310 \pm 0.055$ (+0.274) | $0.952 \pm 0.009$ (+0.140) | $0.468 \pm 0.013$ (-1.293) |
| IHDP | CTGAN | $0.663 \pm 0.018$ | $0.419 \pm 0.013$ | $0.888 \pm 0.010$ | $2.521 \pm 0.161$ |
| | STEAM $_{\text{CTGAN}}$ | $0.674 \pm 0.014$ (+0.011) | $0.424 \pm 0.011$ (+0.005) | $0.928 \pm 0.009$ (+0.040) | $1.709 \pm 0.052$ (-0.812) |
| | TabDDPM | $0.477 \pm 0.036$ | $0.340 \pm 0.022$ | $0.862 \pm 0.011$ | $2.706 \pm 0.138$ |
| | STEAM $_{\text{TabDDPM}}$ | $0.553 \pm 0.029$ (+0.076) | $0.396 \pm 0.015$ (+0.056) | $0.918 \pm 0.011$ (+0.056) | $2.346 \pm 0.088$ (-0.360) |
| | ARF | $0.528 \pm 0.009$ | $0.381 \pm 0.010$ | $0.921 \pm 0.009$ | $3.019 \pm 0.117$ |
| | STEAM $_{\text{ARF}}$ | $0.565 \pm 0.014$ (+0.037) | $0.394 \pm 0.010$ (+0.013) | $0.921 \pm 0.009$ (+0.000) | $1.629 \pm 0.056$ (-1.390) |
| | TVAE | $0.622 \pm 0.014$ | $0.410 \pm 0.010$ | $0.880 \pm 0.014$ | $3.198 \pm 0.172$ |
| | STEAM $_{\text{TVAE}}$ | $0.629 \pm 0.015$ (+0.007) | $0.412 \pm 0.011$ (+0.002) | $0.927 \pm 0.007$ (+0.047) | $2.100 \pm 0.075$ (-1.098) |
| | NFlow | $0.406 \pm 0.028$ | $0.309 \pm 0.012$ | $0.882 \pm 0.012$ | $3.835 \pm 0.345$ |
| | STEAM $_{\text{NFlow}}$ | $0.435 \pm 0.034$ (+0.029) | $0.333 \pm 0.020$ (+0.024) | $0.921 \pm 0.007$ (+0.039) | $2.177 \pm 0.118$ (-1.658) |
| ACIC | TVAE | $0.901 \pm 0.014$ | $0.513 \pm 0.004$ | $0.929 \pm 0.005$ | $4.223 \pm 0.138$ |
| | STEAM $_{\text{TVAE}}$ | $0.900 \pm 0.014$ (-0.001) | $0.514 \pm 0.004$ (+0.001) | $0.972 \pm 0.002$ (+0.043) | $2.422 \pm 0.118$ (-1.801) |
| | CTGAN | $0.880 \pm 0.016$ | $0.421 \pm 0.013$ | $0.942 \pm 0.005$ | $4.518 \pm 0.186$ |
| | STEAM $_{\text{CTGAN}}$ | $0.873 \pm 0.014$ (-0.007) | $0.424 \pm 0.014$ (+0.003) | $0.972 \pm 0.002$ (+0.030) | $2.268 \pm 0.154$ (-2.250) |
| | ARF | $0.828 \pm 0.003$ | $0.430 \pm 0.002$ | $0.945 \pm 0.002$ | $4.633 \pm 0.146$ |
| | STEAM $_{\text{ARF}}$ | $0.835 \pm 0.004$ (+0.007) | $0.430 \pm 0.004$ (+0.000) | $0.977 \pm 0.002$ (+0.032) | $2.449 \pm 0.149$ (-2.184) |
| | NFlow | $0.748 \pm 0.019$ | $0.333 \pm 0.014$ | $0.838 \pm 0.035$ | $5.068 \pm 0.282$ |
| | STEAM $_{\text{NFlow}}$ | $0.744 \pm 0.021$ (-0.004) | $0.333 \pm 0.010$ (+0.000) | $0.971 \pm 0.002$ (+0.133) | $2.938 \pm 0.149$ (-2.130) |
| | TabDDPM | $0.124 \pm 0.028$ | $0.002 \pm 0.001$ | $0.813 \pm 0.023$ | $9.281 \pm 1.033$ |
| | STEAM $_{\text{TabDDPM}}$ | $0.141 \pm 0.035$ (+0.017) | $0.002 \pm 0.000$ (+0.000) | $0.955 \pm 0.019$ (+0.142) | $4.497 \pm 0.501$ (-4.784) |
| Jobs | TabDDPM | $0.890 \pm 0.014$ | $0.477 \pm 0.011$ | $0.949 \pm 0.004$ | $3.335 \pm 0.516$ |
| | STEAM $_{\text{TabDDPM}}$ | $0.929 \pm 0.009$ (+0.039) | $0.493 \pm 0.008$ (+0.016) | $0.954 \pm 0.003$ (+0.005) | $1.446 \pm 0.052$ (-1.889) |
| | ARF | $0.832 \pm 0.010$ | $0.431 \pm 0.019$ | $0.964 \pm 0.004$ | $3.173 \pm 0.691$ |
| | STEAM $_{\text{ARF}}$ | $0.863 \pm 0.011$ (+0.031) | $0.481 \pm 0.016$ (+0.050) | $0.953 \pm 0.004$ (-0.011) | $2.280 \pm 0.381$ (-0.893) |
| | TVAE | $0.886 \pm 0.017$ | $0.288 \pm 0.009$ | $0.944 \pm 0.006$ | $4.471 \pm 0.336$ |
| | STEAM $_{\text{TVAE}}$ | $0.887 \pm 0.014$ (+0.001) | $0.300 \pm 0.012$ (+0.012) | $0.949 \pm 0.004$ (+0.005) | $1.540 \pm 0.167$ (-2.931) |
| | CTGAN | $0.830 \pm 0.049$ | $0.339 \pm 0.023$ | $0.925 \pm 0.033$ | $4.608 \pm 0.792$ |
| | STEAM $_{\text{CTGAN}}$ | $0.778 \pm 0.076$ (-0.052) | $0.298 \pm 0.030$ (-0.041) | $0.939 \pm 0.007$ (+0.014) | $1.846 \pm 0.270$ (-2.762) |
| | NFlow | $0.716 \pm 0.058$ | $0.374 \pm 0.017$ | $0.920 \pm 0.018$ | $5.445 \pm 0.883$ |
| | STEAM $_{\text{NFlow}}$ | $0.800 \pm 0.041$ (+0.084) | $0.375 \pm 0.017$ (+0.001) | $0.952 \pm 0.006$ (+0.032) | $2.666 \pm 0.200$ (-2.779) |

## J   Ablation study

To add to the evidence of STEAM's efficacy, we conduct an ablative study by assessing how jointly modelling $P_{\mathbf{X},W}$ affects performance. On the medical datasets used in the main body of the paper, we compare performance of the best standard models with their relevant ablation STEAM$_{\diamond, \text{joint X,W}}$, which models $P_{\mathbf{X},W}$ with the generative model and $P_{Y|W,\mathbf{X}}$ with a PO estimator, and regular STEAM. We report the results in Table 13.

We see that the ablative model, while often improving upon standard generation, is not as effective as STEAM. Directly modelling $P_{W|\mathbf{X}}$, as STEAM does, better preserves the treatment assignment and outcome generation mechanisms, and both JSD$_\pi$ and $U_{\text{PEHE}}$ are significantly improved by STEAM in most cases. Using the full inductive bias of directly modelling each distribution of our desiderata, and following the true DGP of data containing treatments is the best approach to generation.

Table 13: $P_{\alpha,\mathbf{X}}$, $R_{\beta,\mathbf{X}}$, JSD$_\pi$, and $U_{\text{PEHE}}$ values on standard, ablation, and STEAM models. Ablation results averaged over 5 runs, with 95% CIs, other results use 20 runs.

| Dataset | Model | $P_{\alpha,\mathbf{X}}$ (↑) | $R_{\beta,\mathbf{X}}$ (↑) | JSD$_\pi$ (↑) | $U_{\text{PEHE}}$ (↓) |
|---|---|---|---|---|---|
| ACTG | TVAE | $0.926 \pm 0.013$ | $0.483 \pm 0.010$ | $0.946 \pm 0.004$ | $0.564 \pm 0.017$ |
| | STEAM $_{\text{TVAE, joint X,W}}$ (*ablation*) | $0.918 \pm 0.021$ | $0.473 \pm 0.012$ | $0.939 \pm 0.010$ | $0.475 \pm 0.012$ |
| | STEAM $_{\text{TVAE}}$ | $0.929 \pm 0.008$ | $0.486 \pm 0.009$ | $0.958 \pm 0.004$ | $0.492 \pm 0.011$ |
| IHDP | CTGAN | $0.663 \pm 0.018$ | $0.419 \pm 0.013$ | $0.888 \pm 0.010$ | $2.521 \pm 0.161$ |
| | STEAM $_{\text{CTGAN, joint X,W}}$ (*ablation*) | $0.639 \pm 0.021$ | $0.428 \pm 0.009$ | $0.908 \pm 0.019$ | $2.140 \pm 0.134$ |
| | STEAM $_{\text{CTGAN}}$ | $0.674 \pm 0.014$ | $0.424 \pm 0.011$ | $0.928 \pm 0.009$ | $1.709 \pm 0.052$ |
| ACIC | TVAE | $0.901 \pm 0.014$ | $0.513 \pm 0.004$ | $0.929 \pm 0.005$ | $4.223 \pm 0.138$ |
| | STEAM $_{\text{TVAE, joint X,W}}$ (*ablation*) | $0.873 \pm 0.022$ | $0.512 \pm 0.010$ | $0.930 \pm 0.019$ | $2.447 \pm 0.249$ |
| | STEAM $_{\text{TVAE}}$ | $0.900 \pm 0.014$ | $0.514 \pm 0.004$ | $0.972 \pm 0.002$ | $2.422 \pm 0.118$ |

### J.1   $Q_{W|\mathbf{X}}$ ablation

We conduct a further ablation study to identify how setting different classifiers as $Q_{W|\mathbf{X}}$ within STEAM models affects performance. On the IHDP dataset, we compare setting $Q_{W|\mathbf{X}}$ with a logistic regression classifier, as is done in the main experimental section, against a random forest classier. We report the respective JSD$_\pi$ scores for each model in Table 14. We can see that there is no significant difference between how each classifier preserves the treatment assignment mechanism in IHDP. Model selection for $Q_{W|\mathbf{X}}$ is a much smaller priority than selecting $Q_{\mathbf{X}}$, which can significantly affect all metrics, as shown in Table 3.

Table 14: JSD$_\pi$ values for STEAM models with different classifiers for $Q_{W|\mathbf{X}}$.

| Dataset | Model | JSD$_\pi$ (↑) |
|---|---|---|
| IHDP | STEAM $_{\text{CTGAN, log.reg.}}$ | $0.928 \pm 0.009$ |
| | STEAM $_{\text{CTGAN, rf.}}$ | $0.919 \pm 0.020$ |
| | STEAM $_{\text{TabDDPM, log.reg.}}$ | $0.918 \pm 0.011$ |
| | STEAM $_{\text{TabDDPM, rf.}}$ | $0.931 \pm 0.015$ |
| | STEAM $_{\text{ARF, log.reg.}}$ | $0.921 \pm 0.009$ |
| | STEAM $_{\text{ARF, rf.}}$ | $0.922 \pm 0.016$ |
| | STEAM $_{\text{TVAE, log.reg.}}$ | $0.927 \pm 0.007$ |
| | STEAM $_{\text{TVAE, rf.}}$ | $0.915 \pm 0.020$ |
| | STEAM $_{\text{NFlow, log.reg.}}$ | $0.921 \pm 0.007$ |
| | STEAM $_{\text{NFlow, rf.}}$ | $0.924 \pm 0.016$ |

# K Hyperparameter stability

The performance of generative models is typically sensitive to hyperparameters. To assess the stability of STEAM's performance across hyperparameters, on IHDP, we compare CTGAN with STEAM$_{\text{CTGAN}}$ with multiple hyperparameter configurations. We report results by changing three hyperparameters: number of hidden units within the generator layers (`generator_n_units_hidden`) (Table 15), number of hidden layers within the generator (`generator_n_layers_hidden`) (Table 16), and activation functions used in the generator (`generator_nonlin`) (Table 17), keeping all other hyperparameters default.

The performance gap between STEAM$_{\text{CTGAN}}$ and CTGAN is relatively stable across these configurations. STEAM$_{\text{CTGAN}}$ outperforms CTGAN in each metric at almost all hyperparameter levels. The most statistically significant differences are consistently noted in the JSD$_\pi$ and $U_{\text{PEHE}}$ metrics, which is compatible with the results displayed in the main paper.

Table 15: Comparison of STEAM with standard generation on IHDP at different `generator_n_units_hidden` levels. Averaged over 5 runs, with 95% CIs.

| generator_n_units_hidden | Model | $P_{\alpha,X}$ (↑) | $R_{\beta,X}$ (↑) | JSD$_\pi$ (↑) | $U_{\text{PEHE}}$ (↓) |
|---|---|---|---|---|---|
| 5 | CTGAN | $0.517 \pm 0.026$ | $0.396 \pm 0.015$ | $0.863 \pm 0.033$ | $2.914 \pm 0.047$ |
| | STEAM$_{\text{CTGAN}}$ | $0.565 \pm 0.011$ | $0.405 \pm 0.011$ | $0.941 \pm 0.000$ | $2.194 \pm 0.265$ |
| 50 | CTGAN | $0.622 \pm 0.028$ | $0.411 \pm 0.043$ | $0.916 \pm 0.15$ | $2.282 \pm 0.141$ |
| | STEAM$_{\text{CTGAN}}$ | $0.664 \pm 0.020$ | $0.444 \pm 0.017$ | $0.905 \pm 0.041$ | $1.960 \pm 0.174$ |
| 100 | CTGAN | $0.607 \pm 0.038$ | $0.418 \pm 0.032$ | $0.894 \pm 0.010$ | $2.560 \pm 0.289$ |
| | STEAM$_{\text{CTGAN}}$ | $0.682 \pm 0.016$ | $0.439 \pm 0.018$ | $0.912 \pm 0.004$ | $2.097 \pm 0.095$ |
| 300 | CTGAN | $0.619 \pm 0.030$ | $0.434 \pm 0.030$ | $0.908 \pm 0.023$ | $2.426 \pm 0.289$ |
| | STEAM$_{\text{CTGAN}}$ | $0.699 \pm 0.018$ | $0.458 \pm 0.015$ | $0.928 \pm 0.016$ | $2.028 \pm 0.163$ |
| 500 | CTGAN | $0.663 \pm 0.018$ | $0.419 \pm 0.013$ | $0.888 \pm 0.010$ | $2.521 \pm 0.161$ |
| | STEAM$_{\text{CTGAN}}$ | $0.674 \pm 0.014$ | $0.424 \pm 0.011$ | $0.928 \pm 0.009$ | $1.709 \pm 0.052$ |

Table 16: Comparison of STEAM with standard generation on IHDP at different `generator_n_layers_hidden` levels. Averaged over 5 runs, with 95% CIs.

| generator_n_layers_hidden | Model | $P_{\alpha,X}$ (↑) | $R_{\beta,X}$ (↑) | JSD$_\pi$ (↑) | $U_{\text{PEHE}}$ (↓) |
|---|---|---|---|---|---|
| 2 | CTGAN | $0.663 \pm 0.018$ | $0.419 \pm 0.013$ | $0.888 \pm 0.010$ | $2.521 \pm 0.161$ |
| | STEAM$_{\text{CTGAN}}$ | $0.674 \pm 0.014$ | $0.424 \pm 0.011$ | $0.928 \pm 0.009$ | $1.709 \pm 0.052$ |
| 3 | CTGAN | $0.595 \pm 0.067$ | $0.395 \pm 0.066$ | $0.868 \pm 0.064$ | $2.982 \pm 0.647$ |
| | STEAM$_{\text{CTGAN}}$ | $0.693 \pm 0.075$ | $0.441 \pm 0.043$ | $0.924 \pm 0.018$ | $2.028 \pm 0.143$ |
| 4 | CTGAN | $0.583 \pm 0.049$ | $0.259 \pm 0.074$ | $0.807 \pm 0.054$ | $3.278 \pm 0.191$ |
| | STEAM$_{\text{CTGAN}}$ | $0.596 \pm 0.220$ | $0.301 \pm 0.084$ | $0.886 \pm 0.014$ | $2.690 \pm 0.836$ |
| 5 | CTGAN | $0.490 \pm 0.092$ | $0.313 \pm 0.069$ | $0.770 \pm 0.127$ | $2.871 \pm 0.599$ |
| | STEAM$_{\text{CTGAN}}$ | $0.691 \pm 0.071$ | $0.386 \pm 0.041$ | $0.915 \pm 0.010$ | $2.498 \pm 0.536$ |

Table 17: Comparison of STEAM with standard generation on IHDP at different `generator_nonlin` settings. Averaged over 5 runs, with 95% CIs.

| generator_nonlin | Model | $P_{\alpha,X}$ (↑) | $R_{\beta,X}$ (↑) | JSD$_\pi$ (↑) | $U_{\text{PEHE}}$ (↓) |
|---|---|---|---|---|---|
| ReLU | CTGAN | $0.663 \pm 0.018$ | $0.419 \pm 0.013$ | $0.888 \pm 0.010$ | $2.521 \pm 0.161$ |
| | STEAM$_{\text{CTGAN}}$ | $0.674 \pm 0.014$ | $0.424 \pm 0.011$ | $0.928 \pm 0.009$ | $1.709 \pm 0.052$ |
| SELU | CTGAN | $0.604 \pm 0.020$ | $0.419 \pm 0.015$ | $0.855 \pm 0.023$ | $2.509 \pm 0.160$ |
| | STEAM$_{\text{CTGAN}}$ | $0.699 \pm 0.017$ | $0.445 \pm 0.025$ | $0.929 \pm 0.014$ | $2.043 \pm 0.130$ |
| Leaky ReLU | CTGAN | $0.648 \pm 0.045$ | $0.415 \pm 0.015$ | $0.889 \pm 0.016$ | $2.482 \pm 0.210$ |
| | STEAM$_{\text{CTGAN}}$ | $0.699 \pm 0.028$ | $0.457 \pm 0.019$ | $0.916 \pm 0.011$ | $2.036 \pm 0.135$ |

# L  Congeniality bias

Congeniality bias [16] is a phenomenon that may arise from generation with STEAM. In this scenario, it refers to the fact that downstream models which are structurally similar to the outcome generator, $Q_{Y|W,\mathbf{X}}$, may be advantaged in their performance on $\mathcal{D}_s$. For example, if the potential outcomes from an S-learner are used for $Q_{Y|W,\mathbf{X}}$, the outcome generation mechanism in $\mathcal{D}_s$ may be modelled in such a way that it allows downstream S-learners to better estimate CATEs than other learners. While we acknowledge this phenomenon may disadvantage certain downstream models, we note that our outcome error metric, $U_{\text{PEHE}}$, averages across a number of downstream learner types, such that conducting generative model selection with $U_{\text{PEHE}}$ should lead to good performance across a wide variety of downstream learners, not just those similar to $Q_{Y|W,\mathbf{X}}$, helping to reduce this congeniality bias.

## M Causal generative model comparison

For the baseline models in §7.1.2, we use the code provided by [12] in their GitHub `https://github.com/patrickrchao/DiffusionBasedCausalModels`, and we use the same hyperparameter settings for both ANM and DCM as in that work.

In Table 18 we report the full set of results for ANM and DCM with each of these graph discovery methods. We see that the differences between the graph discovery methods are relatively small, except on the IHDP dataset, where $\mathcal{G}_{\text{discovered}}$ is significantly better than $\mathcal{G}_{\text{naive}}$ for the DCM model.

Table 18: $P_{\alpha,\mathbf{X}}$, $R_{\beta,\mathbf{X}}$, $\text{JSD}_\pi$, and $U_{\text{PEHE}}$ values for CGMs with different graph discovery methods. Averaged over 20 runs, with 95% confidence intervals.

| Dataset | Model | $P_{\alpha,\mathbf{X}}$ ($\uparrow$) | $R_{\beta,\mathbf{X}}$ ($\uparrow$) | $\text{JSD}_\pi$ ($\uparrow$) | $U_{\text{PEHE}}$ ($\downarrow$) |
|---|---|---|---|---|---|
| ACTG | DCM $\mathcal{G}_{\text{naive}}$ | $0.773 \pm 0.013$ | $0.369 \pm 0.006$ | $0.937 \pm 0.006$ | $0.665 \pm 0.034$ |
| | DCM $\mathcal{G}_{\text{discovered}}$ | $0.756 \pm 0.011$ | $0.350 \pm 0.007$ | $0.956 \pm 0.005$ | $0.605 \pm 0.023$ |
| | DCM $\mathcal{G}_{\text{pruned}}$ | $0.758 \pm 0.013$ | $0.358 \pm 0.007$ | $0.957 \pm 0.003$ | $0.596 \pm 0.017$ |
| | ANM $\mathcal{G}_{\text{naive}}$ | $0.787 \pm 0.007$ | $0.389 \pm 0.008$ | $0.954 \pm 0.005$ | $0.580 \pm 0.017$ |
| | ANM $\mathcal{G}_{\text{discovered}}$ | $0.836 \pm 0.007$ | $0.419 \pm 0.007$ | $0.952 \pm 0.004$ | $0.578 \pm 0.019$ |
| | ANM $\mathcal{G}_{\text{pruned}}$ | $0.839 \pm 0.008$ | $0.412 \pm 0.005$ | $0.952 \pm 0.005$ | $0.582 \pm 0.014$ |
| IHDP* | DCM $\mathcal{G}_{\text{naive}}$ | $0.557 \pm 0.010$ | $0.340 \pm 0.009$ | $0.883 \pm 0.016$ | $4.878 \pm 0.395$ |
| | DCM $\mathcal{G}_{\text{discovered}}$ | $0.658 \pm 0.011$ | $0.360 \pm 0.007$ | $0.893 \pm 0.008$ | $2.059 \pm 0.140$ |
| | ANM $\mathcal{G}_{\text{naive}}$ | $0.597 \pm 0.029$ | $0.379 \pm 0.011$ | $0.900 \pm 0.005$ | $1.868 \pm 0.147$ |
| | ANM $\mathcal{G}_{\text{discovered}}$ | $0.589 \pm 0.012$ | $0.359 \pm 0.009$ | $0.892 \pm 0.008$ | $1.865 \pm 0.059$ |
| ACIC† | DCM $\mathcal{G}_{\text{discovered}}$ | $0.942 \pm 0.004$ | $0.422 \pm 0.003$ | $0.957 \pm 0.003$ | $4.249 \pm 0.132$ |
| | DCM $\mathcal{G}_{\text{pruned}}$ | $0.939 \pm 0.004$ | $0.420 \pm 0.004$ | $0.959 \pm 0.002$ | $4.340 \pm 0.159$ |
| | ANM $\mathcal{G}_{\text{discovered}}$ | $0.929 \pm 0.003$ | $0.404 \pm 0.003$ | $0.872 \pm 0.002$ | $4.193 \pm 0.127$ |
| | ANM $\mathcal{G}_{\text{pruned}}$ | $0.930 \pm 0.004$ | $0.404 \pm 0.003$ | $0.880 \pm 0.002$ | $4.481 \pm 0.174$ |

* $\mathcal{G}_{\text{pruned}}$ was the same as $\mathcal{G}_{\text{discovered}}$ for IHDP

† Excessive runtime caused the exclusion of $\mathcal{G}_{\text{naive}}$ ACIC results

# N Differential privacy with STEAM

Theoretical guarantees of the privacy of synthetic data are often required in high-stakes scenarios, such as medicine. STEAM can permit this, satisfying DP when its three component models do, as an application of the post-processing and composition theorems of DP [23].

**Proposition 1.** *If $Q_{\mathbf{X}}$, $Q_{W|\mathbf{X}}$, and $Q_{Y|W,\mathbf{X}}$ satisfy $(\epsilon_{\mathbf{X}}, \delta_{\mathbf{X}})$-, $(\epsilon_W, \delta_W)$-, and $(\epsilon_Y, \delta_Y)$-differential privacy respectively, STEAM satisfies $(\epsilon_{total}, \delta_{total})$-differential privacy, where $\epsilon_{total} = \epsilon_{\mathbf{X}} + \epsilon_W + \epsilon_Y$, $\delta_{total} = \delta_{\mathbf{X}} + \delta_W + \delta_Y$.*

*Proof.* See below. □

There are a number of existing DP generative models, classifiers, and regressors which can be set as $Q_{\mathbf{X}}$, $Q_{W|\mathbf{X}}$, and $Q_{Y|W,\mathbf{X}}$ respectively to enable this.

## N.1 STEAM differential privacy proof

The theoretical guarantee of STEAM's differential privacy (DP) when using individual DP components is grounded in the post-processing and composition theorems of DP [23], as we state in the main body of the paper. We make this derivation clear here by first outlining the post-processing and composition theorems in full.

**Theorem** (Post-Processing Theorem). *Let $M : \mathbb{N}^{|\mathcal{X}|} \to \mathcal{R}$ be a randomised algorithm that is $(\epsilon, \delta)$-differentially private. Let $f : \mathcal{R} \to \mathcal{R}'$ be an arbitrary randomised mapping. Then the composition $f \circ M : \mathbb{N}^{|\mathcal{X}|} \to \mathcal{R}'$ is $(\epsilon, \delta)$-differentially private.*

**Theorem** (Composition Theorem). *Let $M_i : \mathbb{N}^{|\mathcal{X}|} \to \mathcal{R}_i$ be an $(\epsilon_i, \delta_i)$-differentially private algorithm for $i \in [k]$. Define $M_{[k]} : \mathbb{N}^{|\mathcal{X}|} \to \prod_{i=1}^{k} \mathcal{R}_i$ as:*

$$M_{[k]}(x) = (M_1(x), M_2(x), \dots, M_k(x)),$$

*then $M_{[k]}$ is $\left( \sum_{i=1}^{k} \epsilon_i, \sum_{i=1}^{k} \delta_i \right)$-differentially private.*

Given these theorems, we have our guarantee of DP generation with STEAM, as stated in Proposition 1. Specifically:

*Proof.* $Q_{\mathbf{X}}$ generates $\mathbf{X}$, and satisfies $(\epsilon_{\mathbf{X}}, \delta_{\mathbf{X}})$-differential privacy by assumption.

By the post-processing theorem, inputting $\mathbf{X}$ as the condition to $Q_{W|\mathbf{X}}$ does not affect its privacy. $Q_{W|\mathbf{X}}$ generates $W$, and satisfies $(\epsilon_W, \delta_W)$-differential privacy by assumption.

By the post-processing theorem, inputting $W$ and $\mathbf{X}$ as the conditions to $Q_{Y|W,\mathbf{X}}$ does not affect their privacy. $Q_{Y|W,\mathbf{X}}$ generates $Y$, and satisfies $(\epsilon_Y, \delta_Y)$-differential privacy by assumption.

STEAM generates $(\mathbf{X}, W, Y)$, and is the composition of $Q_{\mathbf{X}}$, $Q_{W|\mathbf{X}}$, and $Q_{Y|W,\mathbf{X}}$, i.e. STEAM = $(Q_{\mathbf{X}}, Q_{W|\mathbf{X}}, Q_{Y|W,\mathbf{X}})$

Therefore, by the composition theorem STEAM satisfies $(\epsilon_{\text{total}}, \delta_{\text{total}})$-differential privacy, where $\epsilon_{\text{total}} = \epsilon_{\mathbf{X}} + \epsilon_W + \epsilon_Y$, $\delta_{\text{total}} = \delta_{\mathbf{X}} + \delta_W + \delta_Y$. □

## N.2 Differential privacy experiments

We now examine STEAM's performance in $(\epsilon, \delta)$-DP generation. For comparison we use a state-of-the-art DP generative model, AIM [71], set at privacy level $(\epsilon, \delta)$, and, for STEAM, we set $Q_{\mathbf{X}}$ as $(\epsilon/3, \delta/3)$-AIM, $Q_{W|\mathbf{X}}$ as an $(\epsilon/3, \delta/3)$-DP random forest, and $Q_{Y|W,\mathbf{X}}$ as an $(\epsilon/3, \delta/3)$-T-Learner, such that STEAM is also $(\epsilon, \delta)$-DP. We compare performance on the ACTG dataset across $\epsilon \in \{0.25, 0.5, 1, 2, 3, 5, 10, 15\}$ with $\delta = 10^{-6}$ in Figure 5.

> **Takeaway.** STEAM$_{\text{AIM}}$ models $P_{Y|W,\mathbf{X}}$ better on all tested values of $\epsilon$, as $U_{\text{PEHE}}$ is significantly lower than for standard AIM. $P_{W|\mathbf{X}}$ is better modelled by STEAM$_{\text{AIM}}$ at small $\epsilon$, with equivalent performance between the methods at less conservative budgets. $P_{\mathbf{X}}$, on the other hand, is better preserved by standard AIM, scoring higher on $P_{\alpha,\mathbf{X}}$ and $R_{\beta,\mathbf{X}}$ at most $\epsilon$. This is likely because

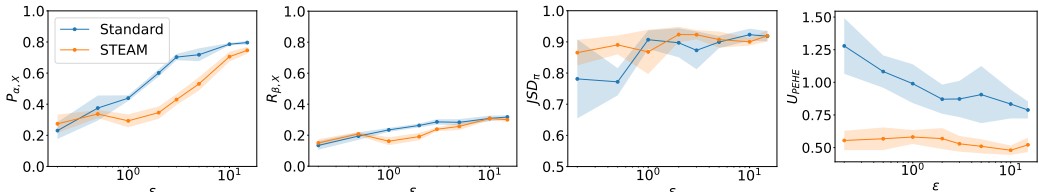

Figure 5: $P_{\alpha,\mathbf{X}}$ (↑), $R_{\beta,\mathbf{X}}$ (↑), $\mathrm{JSD}_\pi$ (↑), and $U_{\mathrm{PEHE}}$ (↓) evaluating STEAM$_{\mathrm{AIM}}$ and standard AIM across privacy budgets. Averaged over 5 runs, shaded area represents 95% CIs.

assigning $Q_{\mathbf{X}}$ one third of the budget of the standard AIM model and having it model largely the same distribution, save for the removed $W$ and $Y$, is prohibitively restrictive given the high-dimensionality of $\mathbf{X}$. As such, with uniform distribution of $(\epsilon, \delta)$ across each component, there is a trade-off between STEAM$_{\mathrm{AIM}}$ and standard AIM, where STEAM$_{\mathrm{AIM}}$ better preserves $P_{W|\mathbf{X}}$ and $P_{Y|W,\mathbf{X}}$, while standard AIM preserves $P_{\mathbf{X}}$ better. Distributing $(\epsilon, \delta)$ differently amongst $Q_{\mathbf{X}}$, $Q_{W|\mathbf{X}}$, and $Q_{Y|W,\mathbf{X}}$ could address this trade-off, as we discuss in Appendix N.4.

### N.3 Extended differential privacy results

We now report $(\epsilon, \delta)$-DP generation results on the ACTG across a wider set of baseline models. In Figure 6 we compare GEM [66] with STEAM$_{\mathrm{GEM}}$, in Figure 7 we compare MST [70] with STEAM$_{\mathrm{MST}}$, and in Figure 8 we compare RAP [5] with STEAM$_{\mathrm{RAP}}$. While there are some nuances to each baseline comparison, the general takeaway remains similar: STEAM models preserve $P_{W|\mathbf{X}}$ and $P_{Y|W,\mathbf{X}}$ better, while standard models preserve $P_{\mathbf{X}}$ better.

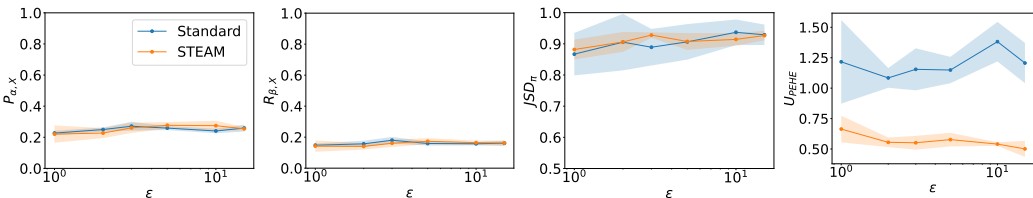

Figure 6: $P_{\alpha,\mathbf{X}}$ (↑), $R_{\beta,\mathbf{X}}$ (↑), $\mathrm{JSD}_\pi$ (↑), and $U_{\mathrm{PEHE}}$ (↓) evaluating STEAM$_{\mathrm{GEM}}$ and standard GEM across privacy budgets. Averaged over 5 runs, shaded area represents 95% CIs.

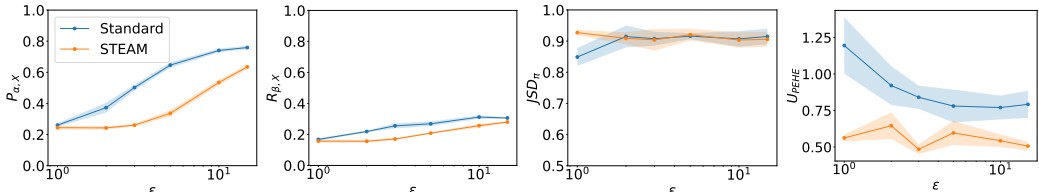

Figure 7: $P_{\alpha,\mathbf{X}}$ (↑), $R_{\beta,\mathbf{X}}$ (↑), $\mathrm{JSD}_\pi$ (↑), and $U_{\mathrm{PEHE}}$ (↓) evaluating STEAM$_{\mathrm{MST}}$ and standard MST across privacy budgets. Averaged over 5 runs, shaded area represents 95% CIs.

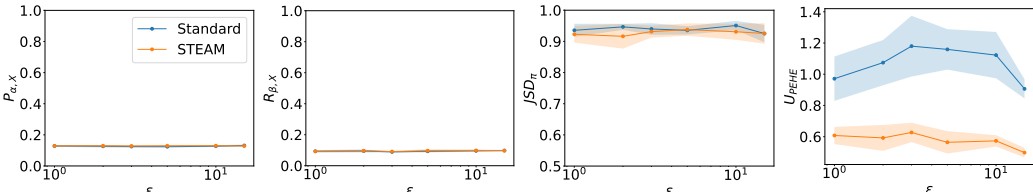

Figure 8: $P_{\alpha,\mathbf{X}}$ (↑), $R_{\beta,\mathbf{X}}$ (↑), $\mathrm{JSD}_\pi$ (↑), and $U_{\mathrm{PEHE}}$ (↓) evaluating STEAM$_{\mathrm{RAP}}$ and standard RAP across privacy budgets. Averaged over 5 runs, shaded area represents 95% CIs.

It is worth noting, however, that when baseline models perform poorly in modelling $P_{\mathbf{X}}$, as is the case for GEM and RAP, then the relevant STEAM model exhibits similar performance in this regard.

### N.4 Allocation of the privacy budget in STEAM

In STEAM, uniform distribution of the privacy budget $\epsilon$ amongst the three component models ensures $(\epsilon, \delta)$-DP. However, such allocation is uninformed on the difficulty of modelling of $P_{\mathbf{X}}$, $P_{W|\mathbf{X}}$, and $P_{Y|W,\mathbf{X}}$, and their relative importance to downstream analysts.

In relation to the importance of each distribution, one immediate improvement can be to distribute $\epsilon$ according to some preference function $f : (0, \infty) \times \triangle^2 \to \epsilon \cdot \triangle^2$ (where $\triangle^2$ is the 2-simplex) which takes input of the budget $\epsilon$ and weights $\mathbf{w}$ for the relative importance of good modelling in $Q_{\mathbf{X}}$, $Q_{W|\mathbf{X}}$, and $Q_{Y|W,\mathbf{X}}$, and outputs a corresponding $\epsilon$ distribution. For example, a simple preference function definition would be $f(\epsilon, \mathbf{w}) = \epsilon \cdot \mathbf{w}$ where $\mathbf{w}$ could be defined by a data holder with some prior knowledge of the importance level of each component distribution to downstream analysts. Another approach, if it is not necessary to specify the desired $\epsilon$ distribution *a priori*, is to treat it as a hyperparameter, to be tuned over a series of runs to optimize some metric, such as a combination of $P_{\alpha,X}$, $R_{\beta,X}$, $\text{JSD}_\pi$, and $U_{\text{PEHE}}$.

Incorporating knowledge of the complexity of modelling $P_{\mathbf{X}}$, $P_{W|\mathbf{X}}$, and $P_{Y|W,\mathbf{X}}$ is more difficult. While some proxy measures could be established, such as the number of covariates in $\mathbf{X}$ indicating the complexity of $P_{W|\mathbf{X}}$, establishing a robust understanding of how the complexity of these distributions relate and compare, is highly non-trivial, and as such, we leave this for future work.

