# OpenReview forum: "Improving the Generation and Evaluation of Synthetic Data for Downstream Medical Causal Inference"
_NeurIPS.cc/2025/Conference — NeurIPS 2025 poster_

### Official Review · Reviewer_SPcu · 2025-06-29

**Clarity:** 4
**Significance:** 2
**Originality:** 4
**Rating:** 5
**Confidence:** 3

**Summary:**

The authors proposed a new framework (metrics + method) for evaluating the fidelity of an estimated distribution using synthetic data in the contexts of treatment data. In many scenarios in the real world (for example healthcare) the data is precisely treatment data, that is, a tuple (Y,X,W) \sim P where Y is the outcome, X the feature vector and W is an ascertainment binary variable that models whether or not a particular treatment was applied.

The authors bring to attention the problem that P is estimated via some marginals using a mix of real and synthetic data. Synthetic data is very common in the settings the paper goes over because not only data is scarce but the questions are counterfactual in nature, real data only has a single outcome observed.  Then they proceed to expose sufficient and necessary desiderata to guarantee the fidelity of the estimated marginals Q and propose metrics that guarantee so. In particular they make the point that the standard methods are based on KL divergence and they prove that in high dimensions the error of such estimates is the same for infinitely many estimates.

Then the authors proceed to introduce STEAM, a framework that performs better under the proposed metrics than standard approachs to estimate Q.

**Questions:**

- Why do we only care about the support as the justification for the metrics for the coovariate distribution? what about the Kurtosis?
- It is not clear to me why  boundedness is part of the desiderata for the distance.
- Is there a result analogous to Theorem 1 for the proposed metrics? If such a result exists, some of the arguments currently used to motivate the new metrics may be redundant. Additionally, is Theorem 1 truly a failure mode specific to KL-based estimations? It seems natural for a distance function to have infinitely many points at a fixed distance from the center—for example, all points on the surface of a ball of fixed radius.

**Ethical Concerns:**

["NO or VERY MINOR ethics concerns only"]

**Limitations:**

Yes

**Quality:**

4

**Strengths And Weaknesses:**

Weakness
- Some of the metrics proposed but the authors are based on unmentioned metrics just mentioned as aa citation, thus not providing neither motivation and making the reading of the paper more difficult. (pg 5)
- The choice of the Jensen-Shannon distance is not justified in the paper, the authors claim symmetry, smoothness, and
boundedness is enough desiderata for this but for me is not self-evident that makes this distance the best way to compare Bernoulli distributions
- The claim "Assessment should therefore centre on a complex task, in which comparable performance will likely imply the same for simpler tasks" is not self evident to me.
Strengths
- The proposal of metrics that do not depend on oracle value and thus do not depend on semisynthetic benchmarks.
- Comparing performance between simulated data against real data as the way to measure the fidelity of the outcome mechanism is a very interesting idea.
- The empirical analysis was clear and well thought as a fair benchmark between the standard methods and STEAM, this is relevant as they are evaluating on their own proposed metrics.

---

> ### Author Rebuttal · Authors · 2025-07-29
>
> Thank you for your thoughtful comments and suggestions. We give answers to each of the following in turn:
>
> - (A) Covariate metrics
> - (B) Jensen-Shannon distance
> - (C) Using a complex task for outcome evaluation
> - (D) Theorem 1
>
> ---
>
> **(A) Covariate metrics**
>
> We are happy to clarify our motivation for choosing $\alpha$-precision and $\beta$-recall as the metrics for covariate evaluation. It is important to note that these metrics do not just compare the supports of the synthetic and real distributions, but they are explicitly _density-aware_, as they examine synthetic and real $\alpha$-supports across a range of $\alpha$. Underlying these metrics is the definition of an $\alpha$-support: the minimum volume subset that contains probability mass $\alpha$. By evaluating how much synthetic data lies in the  $\alpha$-support of the real data, and vice versa, for $\alpha \in [0,1]$, we can obtain a precision-recall curve which, when compared to the ideal, diagonal line achieved only by identical distributions, results in the numerical integrated $\alpha$-precision and $\beta$-recall scores used for model evaluation. These metrics offer a more comprehensive evaluation of distributional alignment than simple support comparison, or comparison of a single moment like kurtosis.
>
> Intuitively, $\alpha$-precision measures the _fidelity_ of the synthetic data, where a high score indicates that the model is not generating unrealistic, low-density samples. $\beta$-recall, on the other hand, measures synthetic data diversity, where a high score indicates that the model has successfully generated samples that cover all the major modes of the real distribution.
>
> We choose these metrics because they allow disentangling of two failure modes—generating unrealistic samples, and mode collapse—which unidimensional metrics, such as KL divergence, do not. Furthermore, they are popular choices in general synthetic data literature [1,2,3,4,5,6].
>
> **Update:** We will **expand Section 5.2.1** to include a more **in-depth explanation of these metrics**.
>
> ---
>
> **(B) Jensen-Shannon distance**
>
> There are many potential distances and divergences that could be used to compare $P_{W|X}$ and $Q_{W|X}$, and, while we choose Jensen-Shannon distance (JSD) in our manuscript, it is by no means that only reasonable choice. On our desire for boundedness, specifically, we see that a key practical advantage of a bounded distance metric is its interpretability. Since JSD is bounded between [0,1], it allows for more immediate, intuitive interpretation of results than unbounded metrics, such as KL divergence, as it is quite clear what is a 'good' and 'bad' score, without requiring further context, such as comparison between different proposed generative models. Given its boundedness, if $JSD_\pi$ is close to 1, we can be sure that the treatment assignment mechanism is modelled well, and vice versa if it is close to 0. With unbounded metrics, such interpretation can be more difficult, as it can be unclear how big of an (e.g. KL) score is to be deemed 'bad'. Indeed, we enjoy the fact that our metrics $P_{\alpha, X}$ and $R_{\beta, X}$ are also bounded in $[0,1]$, allowing similar ease of interpretation.
>
> Despite this, we acknowledge that no metric is universally superior, and quantifying distributional distance is a difficult task in general. We discuss alternatives to JSD in Appendix E. We note that, from our experimental experience, we did not see large differences in the level of information offered by alternative distance metrics, i.e. comparative model rankings tended to remain similar across different distance metrics. Nevertheless, if a completely comprehensive evaluation is desired, it would be wise for a user to perform comparison using a suite of relevant distance metrics.
>
> **Update:** We will **expand Section 5.2.2 to explain our choice of JSD**, while making our discussion on alternatives in Appendix E more clearly referenced.
>
> ---
>
> **(C) Using a complex task for outcome evaluation**
>
> We agree that our original claim, "comparable performance [on a complex task] will likely imply the same for simpler tasks," could be more precise. Our reasoning for this statement is specifically based on the relationship between different causal estimands that are particularly relevant in medical analysis: the Average Treatment Effect (ATE) and the Conditional Average Treatment Effect (CATE). Notably, ATE is the expectation of CATE over the covariate distribution: $ATE = \mathbb{E}_X[CATE(X)]$.
>
> This relationship implies that successfully modelling the entire CATE(X) function is a more comprehensive and stringent test of a synthetic dataset's utility. A dataset that yields accurate CATE estimates across the full patient population will, by definition, also yield an accurate ATE. The reverse is not true; a dataset could reproduce the correct ATE and yet contain incorrect CATEs (e.g., by overestimating the effect for one subgroup and equally underestimating it for another). Therefore, by centering our evaluation on the more complex CATE estimation task, we are performing a more rigorous "stress test" of the outcome generation mechanism $P_{Y|W,X}$. Of course, it is not strictly true that improved CATE preservation in synthetic data will necessarily improve ATE preservation, but it seems a nice heuristic.
>
> **Update:** We will **revise Section 5.2.3 to replace the original statement with this more precise justification** based on the relationship between ATE and CATE.
>
> ---
>
> **(C) Theorem 1**
>
> Our proposed set of metrics is specifically designed to be immune to the failure mode identified in Theorem 1. Since we disentangle the evaluation of the distributions of $X$, $W|X$, and $Y|W,X$, our metrics—$JSD_\pi$ and $U_{PEHE}$ in particular—will maintain _sensitivity_ to the modelling of the treatment assignment and outcome generation mechanisms as the dimensionality of $X$ increases. While Theorem 1 applies to KL divergence (and we see similar empirical behaviour in other joint-level metrics in Appendix D), since we individually evaluate $Q_{W|X}$ and $Q_{Y|W,X}$, our metrics will be able to identify which, amongst a suite of proposal models, best preserves these distributions, even in high-dimensional scenarios. We see this empirically demonstrated in Appendix D, where our metrics allow better identification of modelling differences than any joint-level metric.
>
> We suspect that extensions to Theorem 1 for more joint-level metrics, beyond just KL divergence, are possible. The underlying issue of such metrics is related to the curse of dimensionality, as distances can lose meaning in high dimensions, where all points tend to be equally far away from each other, and the contribution of the low-dimensional W and Y components to the overall distance becomes negligible as d increases, causing the metric to "lose sight" of them.
>
> The ball analogy is an interesting intuition, however it is not quite the same as the point proven by Theorem 1. We prove that, in high dimensions, _any_ differences in the modelling of $Q_{W|X}$ or $Q_{Y|W,X}$ will not be detected by KL divergence, as scores for different models will become increasingly similar. This is not quite the same as saying that there are infinitely many potential distributions that result in the same KL divergence (which is also true, and related to the ball metaphor), as we instead are saying that _any_ models that differ only in these specific conditionals will be equidivergent from the real distribution.
>
> ---
>
> Thank you once again. We hope that we have addressed all your comments, and we greatly appreciate your feedback.
>
> ---
>
> [1] Liu, Tennison, et al. "Goggle: Generative modelling for tabular data by learning relational structure." The Eleventh International Conference on Learning Representations. 2023.
>
> [2] Zhang, Hengrui, et al. "Mixed-type tabular data synthesis with score-based diffusion in latent space." arXiv preprint arXiv:2310.09656 (2023).
>
> [3] Truda, Gianluca. "Generating tabular datasets under differential privacy." arXiv preprint arXiv:2308.14784 (2023).
>
> [4] Ramachandranpillai, Resmi, et al. "Bt-GAN: generating fair synthetic healthdata via bias-transforming generative adversarial networks." Journal of Artificial Intelligence Research 79 (2024): 1313-1341.
>
> [5] Ramachandranpillai, Resmi, Md Fahim Sikder, and Fredrik Heintz. "Fair latent deep generative models (fldgms) for syntax-agnostic and fair synthetic data generation." ECAI 2023. IOS Press, 2023. 1938-1945.
>
> [6] Fang, Zhengyu, et al. "Understanding and mitigating memorization in diffusion models for tabular data." arXiv preprint arXiv:2412.11044 (2024).

---

> > ### Comment · Reviewer_SPcu · 2025-08-04
> >
> > I would like to thank the authors for their thorough clarifications. I decide to maintain my score.

---

> > > ### Author Response · Authors · 2025-08-05
> > >
> > > Thank you again for your engagement in the review of our work. We are glad that you have offered a positive assessment, and we are very grateful for your useful comments that will help to improve the manuscript.

---

### Official Review · Reviewer_5zLf · 2025-07-02

**Clarity:** 3
**Significance:** 4
**Originality:** 3
**Rating:** 5
**Confidence:** 3

**Summary:**

This paper addresses a gap for the generation of synthetic causal data. The authors identify that existing synthetic data generation methods and their evaluation metrics are primarily designed for predictive tasks and fail to capture the causal structure required for robust downstream treatment effect analyses. To remedy this, they propose (a) a set of properties that synthetic data for causal inference should preserve, (ii) new evaluation metrics that directly measure how well these desiderata are met, and (iii) a flexible, model-agnostic generation framework (STEAM) that explicitly targets these properties. Empirical results across multiple real and simulated medical datasets show that STEAM outperforms generic methods, especially as data complexity increases.

**Questions:**

- How sensitive is STEAM’s performance to the choice of base generative models, classifiers, and regressors?
- The focus is on binary treatments. Can STEAM naturally extend to multi-valued or continuous treatments, and are there unique challenges for those settings?

**Ethical Concerns:**

["NO or VERY MINOR ethics concerns only"]

**Final Justification:**

I thank the authors for addressing my questions on the discussion/analysis in the paper, STEAM performance on covariate metrics, model sensitivity, and future extensions.

I maintain my score of accept.

**Limitations:**

yes

**Quality:**

4

**Strengths And Weaknesses:**

Strengths
- Requiring the presentation of {covariate distribution, treatment assignment mechanism, and outcome generation mechanism} is not itself a novel realization, but by articulating those as tests for evaluating synthetic datasets is a valuable formalization.
- This work proposes sound metrics for each of those properties. For instance, it employs the metrics from Alaa et al 2022 (as opposed to e.g. KL Divergence) to evaluate the covariate distribution, which is more robust to outliers and reveal different modes of failure (e.g., mode collapse or over-coverage). It uses a different, prediction-based approach to ensure the assignment mechanisms are present.
- The evaluation is thorough and provides clear ablations of how STEAM compares to other models.
- The advantage of STEAM (by designing it with the inductive biases that we want to preserve) becomes more pronounced as the data or the underlying relationships (e.g., treatment assignment, outcome generation) become more complex or high-dimensional.

Weaknesses
- Based on table 3, the STEAM approach doesn’t always seem to improve the covariate metrics (Pα,X, Rβ,X) much over baseline generative models; most gains are in treatment and outcome mechanisms.
- Due to space limitations, the results and discussions sections are shorter than one would like. It largely boils down to a figure, a brief comment, and then the takeaway. There is very little analysis of Table 3.

---

> ### Author Rebuttal · Authors · 2025-07-29
>
> Thank you for your thoughtful comments and suggestions. We give answers to each of the following in turn:
>
> - (A) STEAM performance on covariate metrics
> - (B) Sensitivity to component models in STEAM
> - (C) Extension to multiple or continuous treatments
>
> ---
>
> **(A) STEAM performance on covariate metrics**
>
> This is indeed an important observation from the results in Table 3. While STEAM consistently and substantially improves the preservation of the treatment assignment and outcome generation mechanisms, its impact on covariate preservation, as measured by $P_{\alpha, X}$ and $R_{\beta, X}$, is more modest and, in some cases, it performs worse.
>
> This is in fact an expected outcome of STEAM's design. The core benefit of STEAM is not necessarily to be a better covariate generator, but rather to supply necessary model capacity to the treatment and outcome distributions, which are easily overshadowed in generic generative models that treat all variables equally. While STEAM does still isolate the modelling of the covariate distribution in $Q_X$, giving the generative model used for this component an ostensibly easier task than a generic, joint-level alternative, the modelling of $P_X$ is nearly of equivalent difficulty to modelling $P_{X,W,Y}$ when the number of covariates is high.
>
> The ACIC dataset from Table 3 is a good example of this: with 60 variables, the task for STEAM's $Q_X$ (modelling 58 variables) is not substantially easier than the task for the corresponding generic generative models (modelling 60 variables), and STEAM's comparative performance in terms of $P_{\alpha, X}$ and $R_{\beta, X}$ is worst on this dataset. Contrast this with the datasets with fewer covariates (ACTG, IHDP) where STEAM more consistently shows a (modest) improvement in $P_{\alpha, X}$ and $R_{\beta, X}$.
>
> **Update:** We will **expand our discussion in Section 7.1** to give more **comprehensive analysis of Table 3**, including the above discussion.
>
> ---
>
> **(B) Sensitivity to component models in STEAM**
>
> We can see from Table 3 that STEAM is sensitive to the choice of generative model used for $Q_X$, as our metrics significantly differ between different STEAM configurations on the same dataset. The choice of model for $Q_X$ naturally affects the $P_{\alpha, X}$ and $R_{\beta, X}$ scores, but it also has significant ramifications on $JSD_\pi$ and $U_{PEHE}$, since poor modelling of $X$ leads to inevitably poor modelling of the conditional distributions for $W|X$ and $Y|W,X$.
>
> On the other have, we found that STEAM was much less sensitive to the choice of classifier and regressors used in $Q_{W|X}$ and $Q_{Y|W,X}$, respectively. In general, amongst a family of reasonable choices for these models, we did not identify substantial changes in the $JSD_\pi$ and $U_{PEHE}$ metrics. This is unsurprising, as modelling these distributions is an easier task than modelling the (usually high dimensional) covariate distribution.
>
> **Update:** We will **expand our discussion in Section 7.1** to include these **takeaways on STEAM's sensitivity to the model choice for $Q_X$**. We will also **add a new ablation** in the appendix examining STEAM's sensitivity to model choices for $Q_{W|X}$ and $Q_{Y|W,X}$.
>
> ---
>
> **(C) Extension to multiple or continuous treatments**
>
> STEAM's modular design does indeed allow for natural extensions beyond the binary treatment setting.
>
> - **Multiple treatments**. STEAM can easily extend to the multiple treatment setting, as a multi-class classifier can be used for $Q_{W|X}$, and PO regressors compatible with $>2$ treatment arms can be used for $Q_{Y|W,X}$. A key practical challenge that can arise here is data sparsity. If some treatment arms have few subjects, which becomes more likely in multiple treatment settings, then certain classes of PO regressors can perform poorly. For example, a T-Learner, which trains separate regressors for each treatment arm using only the subset of training data that receives the relevant treatment, will become impractical when treatment cohorts become too small. This can be combatted by using POs that share some level of representation between treatment groups, e.g. an S-Learner.
> - **Continuous treatments**. More significant architectural changes would be required for this setting. Firstly, $Q_{W|X}$ would have to be replaced by a _regressor_ rather than a classifier. For $Q_{Y|W,X}$, the treatment space could be discretised and then treated like the multiple treatment case above. Alternatively, if we wish to leave the treatment as continuous we could replace the separate PO regressors with methods designed for continuous treatments, such as those based on the generalised propensity score [1].
>
> **Update:** We will **expand Section 8** to include this discussion on **potential extensions of STEAM to multiple and continuous treatment settings**.
>
> ---
>
> Thank you once again. We hope that we have addressed all your comments, and we greatly appreciate your feedback.
>
> ---
> [1] Hirano, K. and Imbens, G.W. (2004). The Propensity Score with Continuous Treatments. In Applied Bayesian Modeling and Causal Inference from Incomplete-Data Perspectives (eds W.A. Shewhart, S.S. Wilks, A. Gelman and X.-L. Meng). https://doi.org/10.1002/0470090456.ch7

---

### Official Review · Reviewer_2zJC · 2025-07-05

**Clarity:** 4
**Significance:** 3
**Originality:** 3
**Rating:** 3
**Confidence:** 3

**Summary:**

This paper addresses the problem of generating and evaluating synthetic data for medical causal inference tasks. The authors identify three core aspects critical for downstream causal analysis: (i) the covariate distribution, (ii) the treatment assignment mechanism, and (iii) the outcome generation mechanism. They propose STEAM, a novel framework that generates synthetic data by explicitly targeting and minimizing errors in each of these aspects. Through experiments on three medical datasets, the authors demonstrate that STEAM consistently outperforms existing generative models, achieving lower error across all three evaluation criteria.

**Questions:**

- It seems to me that the proposed methods can be broadly applicable to causal inference tasks in various domains. However, the paper is positioned specifically within the medical setting. Could the authors clarify what aspects of STEAM are uniquely tailored to medical data, and whether there are domain-specific considerations that guided the design?
- The proposed approach relies on the unconfoundedness assumption, which may not always hold in practice, especially in real-world medical data. Could the authors discuss how STEAM could be adapted or extended to handle settings where this assumption is violated?

**Ethical Concerns:**

["NO or VERY MINOR ethics concerns only"]

**Final Justification:**

I have read the authors' responses and would like to maintain my initial assessment of this work; I think this work can benefit from more revisions based on the discussions and hope my comments are helpful in their future work.

**Limitations:**

Yes

**Quality:**

3

**Strengths And Weaknesses:**

Strengths:
- The paper is exceptionally well-written and easy to follow. The key questions are clearly articulated and the proposed solution is well motivated and explained. I found the paper a pleasure to read.
- The experimental evaluation is comprehensive and demonstrates the effectiveness of the proposed approach across multiple datasets.
- The distinction between STEAM and existing generative methods is clearly stated and well justified.

Weaknesses:
- My main concern is that the proposed approach assumes unconfoundedness, which may not always be realistic in medical settings. It would strengthen the paper to discuss how STEAM might be generalized or adapted when this assumption does not hold.

---

> ### Author Rebuttal · Authors · 2025-07-29
>
> Thank you for your thoughtful comments and suggestions. We give answers to each of the following in turn:
>
> - (A) Confounding
> - (B) Applications beyond medicine
>
> ---
>
> **(A) Confounding**
>
> Thank you for raising this important point. We agree that unconfoundedness is a strong assumption that may not hold in many real-world datasets. We do note, however, that this is a standard assumption in a large body of treatment effect estimation literature [1,2,3,4,5,6,7,8].
>
> Our goal in generating synthetic data, as discussed in Section 4 (lines 116-121), is to create a high-fidelity replica of a real dataset, preserving its characteristics, including any biases and potential confounding. We aim for any synthetic data to be an accurate reflection of real data, rather than seeking improvement and reduction of these biases. This is because we see the problem of detecting, correcting for, and operating under unobserved confounding as orthogonal to the task of synthetic data generation, and indeed there already exist methods for analysis in this setting [9,10]. Such methods can equally be applied when conducting analysis on STEAM-generated data, just like real data.
>
> In fact, one hindrance to improving causal inference methods for complex settings (e.g. those with unobserved confounding) is the lack of freely available datasets that can act as sandboxes for algorithmic development, due to privacy concerns. By preserving real data distributions as accurately as possible, we hope that STEAM-generated data can be used as valuable and safe tools for researchers to develop and validate new methods for analysis under confounding.
>
> That said, we agree that discussing extensions to STEAM for settings where we do not assume unconfounding would strengthen the paper. One extension, for instance, could be valid if an instrumental variable ($Z$) were available, as the STEAM generation process could be adapted to follow the corresponding causal structure:
>
> $Z \sim P_Z, X \sim P_{X|Z}, W \sim P_{W|X,Z}, Y \sim P_{Y|W,X}$.
>
> This would produce synthetic data suitable for downstream analysis using instrumental variable methods, which do not require the unconfoundedness assumption. This represents a potentially fruitful direction for future work. We will add this to our discussion section, to make the assumptions present in STEAM transparent, and detail possible future works to address alternative scenarios.
>
> **Update:** We will **add the above discussion to Section 8** to clarify STEAM's assumptions and goals, while **outlining potential future work to address hidden confounding**.
>
> ---
>
> **(B) Applications beyond medicine**
>
> You are correct that the core principles of STEAM are broadly applicable to causal inference tasks in any domain featuring treatments or interventions. We chose to position the paper within the medical setting for two main reasons:
>
> 1) **Clear practical need.** The medical field faces significant challenges with data access due to necessary privacy regulations, making high-quality synthetic data particularly valuable.
> 2) **Motivating our desiderata.** The medical context helps justify the importance of our specific goals. For example, preserving the covariate distribution $P_X$ is motivated by the standard medical practice of reporting patient cohort statistics, and accurately modeling the treatment assignment mechanism $P_{W|X}$ is shown to be critical given the high stakes of misrepresenting treatment protocols.
>
> Nevertheless, we agree that the STEAM architecture itself, and our proposed evaluation metrics, are domain-agnostic. To underscore this, we have already included an experiment on a non-medical dataset in Appendix H, on the Jobs dataset [11] which contains experimental data from a male sub-sample from the National Supported Work Demonstration used to evaluate the effect of job training on income. The results on this data confirm that STEAM provides the same benefits in this context as in the main body of the paper.
>
> **Update:** We will **expand Section 8 to highlight other domains where STEAM and our metrics can be potentially useful**, and clearly reference to our existing results in Appendix H.
>
> ---
>
> Thank you once again. We hope that we have addressed all your comments, and we greatly appreciate your feedback.
>
> ---
>
> [1] Curth, Alicia, and Mihaela Van der Schaar. "Nonparametric estimation of heterogeneous treatment effects: From theory to learning algorithms." International Conference on Artificial Intelligence and Statistics. PMLR, 2021.
>
> [2] Künzel, Sören R., et al. "Metalearners for estimating heterogeneous treatment effects using machine learning." Proceedings of the national academy of sciences 116.10 (2019): 4156-4165.
>
> [3] Nie, Xinkun, and Stefan Wager. "Quasi-oracle estimation of heterogeneous treatment effects." Biometrika 108.2 (2021): 299-319.
>
> [4] Kennedy, Edward H. "Towards optimal doubly robust estimation of heterogeneous causal effects." Electronic Journal of Statistics 17.2 (2023): 3008-3049.
>
> [5] Wager, Stefan, and Susan Athey. "Estimation and inference of heterogeneous treatment effects using random forests." Journal of the American Statistical Association 113.523 (2018): 1228-1242.
>
> [6] Shalit, Uri, Fredrik D. Johansson, and David Sontag. "Estimating individual treatment effect: generalization bounds and algorithms." International conference on machine learning. PMLR, 2017.
>
> [7] Alaa, Ahmed, and Mihaela Van Der Schaar. "Validating causal inference models via influence functions." International Conference on Machine Learning. PMLR, 2019.
>
> [8] Shi, Claudia, David Blei, and Victor Veitch. "Adapting neural networks for the estimation of treatment effects." Advances in neural information processing systems 32 (2019).
>
> [9] Dennis Frauen and Stefan Feuerriegel. Estimating individual treatment effects under unobserved confounding using binary instruments. arXiv preprint arXiv:2208.08544, 2022.
>
> [10] Nathan Kallus, Xiaojie Mao, and Angela Zhou. Interval estimation of individual-level causal effects under unobserved confounding. In The 22nd international conference on artificial intelligence and statistics, pages 2281–2290. PMLR, 2019.
>
> [11] Robert J LaLonde. Evaluating the econometric evaluations of training programs with experimental data. The American economic review, pages 604–620, 1986.

---

> > ### Comment · Area_Chair_LHzg · 2025-08-07
> > **please respond to rebuttal**
> >
> > Dear Reviewer 2zJC,
> >
> > Could you please let us know whether the rebuttal of the authors sufficiently addresses your concerns?
> >
> > Thank you,
> > The Area Chair

---

> ### Author Response · Authors · 2025-08-06
>
> Dear Reviewer 2zJC,
>
> Thank you again for the helpful comments in your review. We hope that we have addressed all of your concerns with our reply and the suggested changes to our manuscript, and we are more than happy to address any remaining questions you may have before the end of the discussion period. We kindly look forward to your reply.
>
> With thanks,
> The Authors

---

### Official Review · Reviewer_vwba · 2025-07-07

**Clarity:** 4
**Significance:** 2
**Originality:** 3
**Rating:** 5
**Confidence:** 4

**Summary:**

This paper focuses on the availability of medical data for use by the causal evaluation community.  Medical domains are a highly-valued use-case for causal inference methods, but due to patient privacy restrictions, actual medical data is often unavailable.  Synthetic data generation methods exist that approximate the distributions of empirical data.  However, these methods generally treat all the variables in the same way, which is not ideal for causal inference applications, since treatment and outcome have a much greater importance and are used in different ways than the covariates.  The authors propose STEAM, a framework for generating synthetic data that separately generates covariate, treatment, and outcome distributions to match the source data's distributions.  They authors compare STEAM, using different generic synthetic data generation methods for the covariate distribution, against those synthetic data generation methods and find that, in most cases, STEAM improves performance.

**Questions:**

On line 94, you when talking about generic generative models and causal generative models, state that "neither of these existing approaches are tailored to data containing treatment, and they do not target our desiderata".  I agree with respect to generic generative models, but it seems odd to me to claim that "causal generative models" aren't tailored to answering causal questions.  The main issue from my understanding with causal generative models is that they assume you have access to the full causal structure, which is a strong assumption in practice.  In what way are they "not tailed to data containing treatment"?

In addition to the data type assumptions that STEAM is able to make use of (i.e., assuming treatment is binary, so using a Bernoulli for treatment and a logistic regression for outcome), it seems like a major benefit of STEAM is that the treatment and outcome are not having their importance overshadowed by the sheer number of covariates.  Do any of the generic generation methods allow you to adjust the importance of variables, or do any sort of term weighting?  If so, I'd be curious to see if that improves their performance at all.

**Ethical Concerns:**

["NO or VERY MINOR ethics concerns only"]

**Final Justification:**

With the authors agreeing to make the listed changes (cleaning up the narrative and adding the CGM results to the main paper), the paper will only improve. I will maintain my vote for acceptance.

**Limitations:**

I think the limitations are adequately discussed

**Paper Formatting Concerns:**

No formatting concerns

**Quality:**

3

**Strengths And Weaknesses:**

I think this is a strong paper overall.  Given the general difficulty of evaluation in the causal inference space, guidance for how to produce more realistic synthetic data is a welcome contribution.  The authors justify the importance of treatment outcome and treatment separately from covariates, and the flow of the paper overall is clear.  While the idea behind STEAM isn't overly complex (mostly amounting to "model W, X, and Y separately rather than jointly"), it's still a valuable idea to explore, and I think the authors make the case for its practical utility well.  The set of experiments performed against generic generation methods are great and demonstrate the utility of STEAM well.

There's a bit too much repetition in the opening few pages that could be trimmed to make room for, for example, Figure 4 or causal generative model results.  Looking at the first 4 pages of the paper, the authors make some statement that amounts to "generic distribution methods don't treat W (and/or X/Y) separately, which is bad" 9 times from my count (on lines 39, 47, 58, 83, 87, 95, 100, 115, 125).  Obviously, as this is the main thesis of the paper, it's perfectly fine to repeat it.  However, the amount that it's repeated started to become tedious by the end of page 4, since it just felt like the you were repeating yourselves over and over again.  Even removing just a few of these would both save you some space and reduce the repetitive feeling significantly.

On line 197 and 205, the authors mention "Figure 4", which doesn't actually seem to be in the paper.  On further investigation, there is a Figure 4 in Section N of the appendix, but that's not mentioned either time the figure is referenced in the main body of the paper.  It also feels weird to put a diagram showing the core flow of the proposed framework in the appendix.

While the evaluation that is in the paper is good (I like the set up of using each generic generation method for the covariate distribution as a comparison), there are additional evaluations that feel missing.  While I understand that STEAM has the advantage of making fewer assumptions than causal generation methods, I would still be interested in seeing a comparison with them.  The authors are working in a setting where some level of ground truth structure is assumed (covariates are pre-treatment, and treatment is pre-outcome), so from my understanding, the main missing piece for causal generation methods would be not knowing the causal structure among the covariates.  However, I'd still like to see some additional rows in Table 3 for a few different causal generation methods.  It seems like there are a few ways you could do this, such as, treating all covariates as independent (just having them all have arrows into both W and Y), applying a causal structure learning algorithm to the covariates and generating based on that, treating the covariates as fully connected...

---

> ### Author Rebuttal · Authors · 2025-07-29
>
> Thank you for your thoughtful comments and suggestions. We give answers to each of the following in turn:
>
> - (A) Comparison with causal generative models
> - (B) Repetition in writing
> - (C) Adjusting variable importance in generic generative models
>
> ---
>
> **(A) Comparison with causal generative models**
>
> Thank you for highlighting the family of causal generative models (CGMs). We agree that a direct comparison with CGMs is crucial for positioning our work, even though they operate under stronger assumptions than STEAM.
>
> Indeed, we have already conducted a comparison in Appendix L of the best performing STEAM models from Table 3 with two performant CGMs: an additive noise model (ANM) [1] implementation from the _DoWhy-GCM_ python package [2], and a diffusion-based causal model [3]. To fairly apply these methods, we first must reconcile their requirement of a complete causal graph. We investigate three distinct methods for defining the causal graph structure that assume no more knowledge than is encoded in STEAM:
>
> 1) A naive graph where covariates are fully connected, all covariates cause W and Y, and W causes Y ($\mathcal{G}_\text{naive}$);
> 2) A graph learned using the PC causal discovery algorithm [4] ($\mathcal{G}_\text{discovered}$);
> 3) A pruned version of the discovered graph that removes any edges violating the known data generating process, i.e. any backwards edges from Y to W or X, or from W to X are removed ($\mathcal{G}_\text{pruned}$).
>
> We repeat these results from Appendix L below:
>
> | Dataset | Model | $\boldsymbol{P_{\alpha, \mathbf{X}}}$ ($\uparrow$) | $\boldsymbol{R_{\beta, \mathbf{X}}}$ ($\uparrow$) | $\boldsymbol{\textbf{JSD}_\pi}$ ($\uparrow$) | $\boldsymbol{U_\textbf{PEHE}}$ ($\downarrow$) |
> |---|---|---|---|---|---|
> | ACTG | $STEAM_\text{TVAE}$ |**0.929 $\pm$ 0.008**| **0.486 $\pm$ 0.009**| 0.958 $\pm$ 0.004   | **0.492 $\pm$ 0.011**   |
> |  | DCM $\mathcal{G}_\text{naive}$ | 0.773 $\pm$ 0.013 | 0.369 $\pm$ 0.006 | 0.937 $\pm$ 0.006 | 0.665 $\pm$ 0.034 |
> |  | DCM $\mathcal{G}_\text{discovered}$ | 0.756 $\pm$ 0.011 | 0.350 $\pm$ 0.007 | 0.956 $\pm$ 0.005 | 0.605 $\pm$ 0.023 |
> |  | DCM $\mathcal{G}_\text{pruned}$ | 0.758 $\pm$ 0.013 | 0.358 $\pm$ 0.007 | 0.957 $\pm$ 0.003 | 0.596 $\pm$ 0.017 |
> |  | ANM $\mathcal{G}_\text{naive}$ | 0.787 $\pm$ 0.007 | 0.389 $\pm$ 0.008 | 0.954 $\pm$ 0.005 | 0.580 $\pm$ 0.017 |
> |  | ANM $\mathcal{G}_\text{discovered}$ | 0.836 $\pm$ 0.007 | 0.419 $\pm$ 0.007 | 0.952 $\pm$ 0.004 | 0.578 $\pm$ 0.019 |
> |  | ANM $\mathcal{G}_\text{pruned}$ | 0.839 $\pm$ 0.008 | 0.412 $\pm$ 0.005 | 0.952 $\pm$ 0.005 | 0.582 $\pm$ 0.014 |
> |---|---|---|---|---|---|
> | IHDP | $STEAM_\text{CTGAN}$ | 0.674 $\pm$ 0.014  | **0.424 $\pm$ 0.011**| **0.928 $\pm$ 0.009**| **1.709 $\pm$ 0.052** |
> |  | DCM $\mathcal{G}_\text{naive}$ | 0.557 $\pm$ 0.010 | 0.340 $\pm$ 0.009 | 0.883 $\pm$ 0.016 | 4.878 $\pm$ 0.395 |
> |  | DCM $\mathcal{G}_\text{discovered}$ | 0.658 $\pm$ 0.011 | 0.360 $\pm$ 0.007 | 0.893 $\pm$ 0.008 | 2.059 $\pm$ 0.140 |
> |  | DCM ${\mathcal{G}_\text{pruned}}^*$ | 0.658 $\pm$ 0.011 | 0.360 $\pm$ 0.007 | 0.893 $\pm$ 0.008 | 2.059 $\pm$ 0.140 |
> |  | ANM $\mathcal{G}_\text{naive}$ | 0.597 $\pm$ 0.029 | 0.379 $\pm$ 0.011 | 0.900 $\pm$ 0.005 | 1.868 $\pm$ 0.147 |
> |  | ANM $\mathcal{G}_\text{discovered}$ | 0.589 $\pm$ 0.012 | 0.359 $\pm$ 0.009 | 0.892 $\pm$ 0.008 | 1.865 $\pm$ 0.059 |
> |  | ANM ${\mathcal{G}_\text{pruned}}^*$ | 0.589 $\pm$ 0.012 | 0.359 $\pm$ 0.009 | 0.892 $\pm$ 0.008 | 1.865 $\pm$ 0.059 |
> |---|---|---|---|---|---|
> | $\text{ACIC}^\dagger$ | $STEAM_\text{ARF}$ | 0.939 $\pm$ 0.004| 0.393 $\pm$ 0.004 | **0.977 $\pm$ 0.002** | **2.176 $\pm$ 0.141**|
> |  | DCM $\mathcal{G}_\text{discovered}$ | 0.942 $\pm$ 0.004 | 0.422 $\pm$ 0.003 | 0.957 $\pm$ 0.003 | 4.249 $\pm$ 0.132 |
> |  | DCM $\mathcal{G}_\text{pruned}$ | 0.939 $\pm$ 0.004 | 0.420 $\pm$ 0.004 | 0.959 $\pm$ 0.002 | 4.340 $\pm$ 0.159 |
> |  | ANM $\mathcal{G}_\text{discovered}$ | 0.929 $\pm$ 0.003 | 0.404 $\pm$ 0.003 | 0.872 $\pm$ 0.002 | 4.193 $\pm$ 0.127 |
> |  | ANM $\mathcal{G}_\text{pruned}$ | 0.930 $\pm$ 0.004 | 0.404 $\pm$ 0.003 | 0.880 $\pm$ 0.002 | 4.481 $\pm$ 0.174 |
>
> \* $\mathcal{G}$ pruned is the same as $\mathcal{G}$ discovered for IHDP
>
> $\dagger$ Excessive runtime caused the exclusion of $\mathcal{G}_\text{naive}$ ACIC results
>
> The STEAM models outperform these CGM baselines on nearly all metrics and datasets, often to a statistically significant degree, regardless of the underlying causal graph used. STEAM is only outperformed in the ${P_{\alpha, \mathbf{X}}}$ and ${R_{\beta, \mathbf{X}}}$ metrics on the ACIC dataset by the DCM generative model. These results validate our core thesis: when dealing with treatment data, and where the complete causal graph is unknown, STEAM’s encoding of only the high-level, known causal structure leads to more useful and robust synthetic data generation.
>
> Further, to your point on our phrasing in line 94 that CGMs are not "tailored to data containing treatments", we agree that this should be made more precise. Our intention was to highlight that, while CGMs are of course designed with causality in mind and to answer broad sets of causal queries (that require a full causal graph), STEAM is specifically tailored for treatment data. In this setting, the high-level causal structure is known, but finer-grained relationships are typically not. STEAM is designed to leverage precisely this level of knowledge without requiring more restrictive assumptions. We have revised this section in the manuscript to articulate this important distinction more clearly.
>
> **Update:** We will **move the causal generative modelling results** from Appendix L to **Section 7.1**, and **revise line 94** to clarify the differences in assumptions and typical use cases between STEAM and CGMs.
>
> ---
>
> **(B) Repetition in writing**
>
> We appreciate your help in improving the quality of our writing. We agree that we are overly repetitive of our main contention in the first few sections, and we have removed some of the repetitions identified. We use the resulting space to move Figure 4 into the main body of the text.
>
> **Update:** We will **reduce repetitive phrasing** and **move Figure 4 into the main body** of the text.
>
> ---
>
> **(C) Adjusting variable importance in generic generative models**
>
> Thank you for bringing up this interesting potential alternative approach. To our knowledge, there is not substantial existing literature on biasing tabular generative models towards better preserving particular subsets of variables. Nevertheless, it seems like simple modifications could be made to the loss functions of many typical generative models to introduce a weighting vector $w$. For example, a diffusion model could be trained with a weighted noise prediction loss to bias the model to better preserve the treatment and outcome distributions
>
> $\mathcal{L} = \mathbb{E}_{t, x_0, \epsilon} [\sum_d w_d (\epsilon_d - NN(x_t, t)_d)^2]$
>
> where $w_d$ is high for the $W$ and $Y$ variables. Such approaches would require manually tuning these weights, however, and it is not clear what the optimal weights to give to the treatment/outcome variables would be, which is a consideration that is not required in STEAM.
>
> More fundamentally, this weighting scheme would target the _marginal_ distributions of $W$ and $Y$, while we are most interested in their _conditional_ distributions, conditioned on $X$, for downstream tasks like examnination of treatment assignment protocols, and treatment effect estimation. STEAM, on the other hand, specifically targets these conditionals of interest.
>
> ---
>
> Thank you once again. We hope that we have addressed all your comments, and we greatly appreciate your feedback.
>
> ---
>
> [1] Patrik Hoyer, Dominik Janzing, Joris M Mooij, Jonas Peters, and Bernhard Schölkopf. Nonlinear causal discovery with additive noise models. Advances in neural information processing systems, 21, 2008.
>
> [2] Patrick Blöbaum, Peter Götz, Kailash Budhathoki, Atalanti A Mastakouri, and Dominik Janzing. Dowhy-gcm: An extension of dowhy for causal inference in graphical causal models. Journal of Machine Learning Research, 25(147):1–7, 2024.
>
> [3] Patrick Chao, Patrick Blöbaum, Sapan Patel, and Shiva Prasad Kasiviswanathan. Modeling causal mechanisms with diffusion models for interventional and counterfactual queries, 2024.
>
> [4] Peter Spirtes, Clark Glymour, and Richard Scheines. Causation, prediction, and search. MIT press, 2001.

---

> > ### Comment · Reviewer_vwba · 2025-08-05
> >
> > Thank you for your response!  Moving the CGM results into the main paper is a great change.  I think this is a good paper and continue to vote for acceptance.

---

> > > ### Author Response · Authors · 2025-08-05
> > >
> > > Thank you again for your engagement in the review of our work. We are very pleased with your positive assessment, and we are grateful for the useful comments you have provided - they will definitely help improve the manuscript.

---

### Decision · Program_Chairs · 2025-09-17

**Decision:**

Accept (poster)

**Comment:**

The paper introduces a simple method for synthetic data generation that is tailored to causal inference tasks. While positioned in a medical setting, the method is general enough to apply to other settings as well. The paper was well-received by almost all reviewers and the author-reviewer discussion suggested an avenue for minor improvements to make this paper ready for publication. To meet the expectations of the reviewers, the authors should incorporate the rebuttal in their final paper, including improving the clarity of the paper, moving the GCM results to the main paper, and considering the additional baselines brought up by reviewer 2zJC.

An important but overlooked by the reviewers component of this work is the relationship of this work to privacy since synthetically generated data should not leak private information from the original data. I see that the authors present an analysis of their proposed method with respect to differential privacy in the Appendix. I would like to encourage the authors to move the main results on privacy to the main paper. This will provide a more complete picture of the constraints and guarantees of the proposed method.